# Relationship between latent and radiative heating fields of Tropical cloud systems using synergistic satellite observations

Xiaoting Chen[1], Claudia J. Stubenrauch[1], and Giulio Mandorli[1]

[1]Laboratoire de Météorologie Dynamique/Institut Pierre-Simon Laplace, (LMD/IPSL), Sorbonne Université, Ecole Polytechnique, CNRS, Paris, France

**Correspondence:** Xiaoting Chen (xiaoting.chen@lmd.ipsl.fr)

**Abstract.** In order to investigate the relationship between latent and radiative heating (LH, $Q_{rad}$), particularly released by mesoscale convective systems (MCSs), we used synergistic satellite-derived data from active instruments. Given the sparse sampling of these observations, we expanded the Spectral LH profiles derived from the Tropical Rain Measurement Mission (TRMM-SLH) by applying artificial neural network regressions on Clouds from InfraRed Sounder (CIRS) data and meteo-
rological reanalyses, following a similar approach as for the expansion of the $Q_{rad}$ profiles. A direct comparison with the collocated TRMM-SLH data shows an excellent agreement of the average profiles, but the prediction range is underestimated, in particular between 550 and 900 hPa. Noise related to discrepancies in rain fraction between TRMM and CIRS-ML as well as an underestimation of extremes can be reduced by averaging over larger areas. The zonal averages of vertically integrated LH (LP) at 1:30 AM and PM LT align well with those from the full diurnal sampling of TRMM–SLH over ocean. For Upper
Tropospheric (UT) clouds with a large latent heat release, the surface temperature has a larger impact on the atmospheric cloud radiative effect (ACRE) in dry than in humid environments, while the humidity plays a large role in cool than in warm environments. In all situations, the cloud height is mostly responsible for the value of ACRE. The distribution of UT clouds in the LP–ACRE plane shows a very large spread in ACRE for small LP, which is gradually reduced towards larger LP. The mean ACRE of mature MCS increases with LP, up to about 115 W m$^{-2}$. As expected, the shapes of the LH profiles of mature MCSs
show that larger, more organized MCSs have a larger contribution of stratiform rain than the smaller MCSs. Furthermore, convective organization enhances the mean ACRE of mature MCSs on average by about 10 W m$^{-2}$, over the whole LP range.

## 1  Introduction

Clouds envelop approximately two-thirds of the Earth's surface, with 40 % originating from upper tropospheric (UT) clouds (e.g., Stubenrauch et al., 2013, 2017). These cloud formations play a crucial role in modulating the Earth's energy budget and
heat transport. They are most abundant in the tropics, constitute around 60 % of the total tropical cloud distribution and often form as cirrus anvils from convective outflow, building mesoscale convective systems (MCS), as illustrated by Houze (2004).

Throughout the precipitation process, latent heating (LH) is generated within the convective cores and denser regions of the anvils due to the condensation process, which involves the phase transition of water vapor into tiny liquid or frozen cloud particles. The release of latent heat strongly influences the atmospheric circulation, particularly in the tropics (Tao et al., 2006).

Therefore, the interpretation of latent heat release and its fluctuations plays a central role in the complex interactions of the Earth's water and energy cycles (e.g., Gill, 1980; Mapes, 1993; Schumacher et al., 2004; Tao et al., 2016). Radiative heating ($Q_{rad}$) of UT clouds further augments this energy reservoir by at least 20 % (Bergman and Hendon, 2000; Li et al., 2013; Stubenrauch et al., 2021). Given the significance of both processes, this article aims to study the relationship between latent and radiative heating, comparing different environments, with a focus on MCSs.

In order to study the relationship between latent and radiative heating, we have combined information from multiple satellite instruments. The heating profiles are obtained from active measurements which only overlap over a small portion in space and in time with the cloud top properties from passive remote sensing. Previously, we have expanded this vertical information by using machine learning (ML) techniques (Stubenrauch et al., 2021), leading to 3D fields of radiative heating at specific local times (LTs). This article describes the expansion of the latent heating and shows to what extent these 3D latent heating fields at specific LTs represent the daily mean tropical latent heating. In addition, a cloud system approach (Stubenrauch et al., 2023) is

used to identify MCSs and to determine their size, which is related to convective organization (e.g., Houze and Hobbs, 1982; Moncrieff, 1992). The data expansion and the following analyses cover the latitudinal band 30°N-30°S.

  Infrared sounders, with their good spectral resolution, are sensitive to cirrus during both day and night (e.g., Wylie et al., 2005; Stubenrauch et al., 2006, 2017, 2024). For this reason, we use cloud top properties retrieved from measurements of the

Atmospheric Infrared Sounder (AIRS) onboard the National Aeronautics and Space Administration (NASA) Earth Observation Satellite Aqua and of the Infrared Atmospheric Sounding Interferometer (IASI) onboard the EUMETSAT Meteorological Operation satellite (Metop). These instruments are along-track scanners with wide swaths, with a horizontal coverage of about 70 % in the tropics at a specific local time.

  Active sensors, the CALIPSO (Cloud-Aerosol Lidar and Infrared Pathfinder Satellite Observation) lidar and the CloudSat

radar, are part of the A-Train satellite constellation (Stephens et al., 2018) and work in synergy with Aqua since 2006. They provide observations of the vertical structure of clouds (e.g., Mace et al., 2009), radiative heating rates (Henderson et al., 2013), as well as information on precipitation (Haynes et al., 2009). However, these parameters are only available along successive narrow nadir tracks (~2500 km apart).

  The satellite orbit of the Tropical Rainfall Measuring Mission (TRMM) allowed to statistically sample the full diurnal cycle

(Negri et al., 2002). Latent heating profiles have been estimated from the Precipitation Radar (PR) onboard TRMM via the Spectral Latent Heating (SLH) algorithm (Shige et al., 2007). However, within a time window of about one hour, the TRMM data only cover a very small fraction of the tropics of approximately 7 %. For the expansion towards 3D latent heating fields at the specific LTs of AIRS (1:30 AM/PM) and IASI (9:30 AM/PM), we have used a similar approach as in Stubenrauch et al. (2021). While CALIPSO–CloudSat only overlap with AIRS observations, TRMM observations overlap with both, AIRS and

IASI observations.

  Sections 2.1 to 2.3 present the data: latent heating profiles from TRMM PR measurements, cloud properties from AIRS and IASI, 3D structure and precipitation intensity classification from machine learning trained on CloudSat–CALIPSO. Sections 2.4 and 2.5 describe their collocation as well as the training and evaluation of the developed artificial neural network (ANN) models to expand the TRMM–SLH heating profiles. Section 3 outlines the construction of the 3D LH dataset, over the period

2004–2018, and shows its coherence in comparison to the original TRMM-SLH data. Section 4 describes the reconstruction of the MCSs using the CIRS data. Finally, section 5 discusses the results on the relationship between latent and radiative heating, while Section 6 summarizes the key conclusions and suggests future research directions.

In our analyses, we use the following definitions: **LH** refers to the latent heating profile; **LP** denotes the vertically integrated latent heating; $\mathbf{Q}_{rad}$ represents the radiative heating profile; **CRE** (cloud radiative effect) refers to the difference between all-

sky and clear-sky radiative heating rates; and **ACRE** (atmospheric cloud radiative effect) represents the vertically integrated CRE. While the heating rates are expressed in unit of $\mathbf{K\ day^{-1}}$, the vertically integrated heat is given in units of $\mathbf{W\ m^{-2}}$.

## 2 Data, Methods and Evaluation

### 2.1 Latent heating data

The primary objective of the TRMM mission (Houze Jr. et al., 2015; Kummerow et al., 1998; Liu et al., 2012; Dorian, 2014)

was to study the temporal and spatial variability of tropical rainfall. For this purpose, TRMM has an orbital inclination at $35°$ with 16 orbits per day. TRMM revisits a given area at the same LT every 23 (near equator) to 46 days (near $35°$). Therefore, according to (Negri et al., 2002), TRMM–PR estimates over any one hour, even with 3 years of data, are insufficient to accurately describe the diurnal cycle of precipitation for grid sizes smaller than $12°$, due to inconsistent spatial sampling. The PR is a radar operating at the Ku-band in the microwave range around 13.8 GHz, specifically dedicated to obtain vertical

profiles of precipitation, with a horizontal resolution of about 5 km and a swath width of 247 km. The TRMM mission took data from 1997 until 2015 and was then continued by the Global Precipitation Measurement Mission (GPM).

The radar measures the echo-top height, corresponding to the precipitation top height, identifies the melting layer, determines the rain intensity vertical structure, and distinguishes convective and stratiform rain. In general, convective LH profiles show heating throughout the vertical layers, except near the surface due to evaporation at lower levels. LH profiles of the stratiform

rain within the anvils show cooling at low levels below a melting level and heating at levels above (e.g., Chang and L'Ecuyer, 2019, Fig. 3). The peak in LH of isolated convection is also lower in altitude than the one of complex convective systems including stratiform rain in the anvils (e.g., Hartmann et al., 1984).

Since there is no direct measurement of LH, there are two recognized retrieval algorithms to estimate LH for TRMM (Tao et al., 2022): the Convective-Stratiform Heating (CSH, Tao et al., 2018; Lang and Tao, 2018; Tao et al., 2022) and the Spectral

Latent Heating (SLH, Shige et al., 2007, 2008, 2009; Takayabu and Tao, 2020). Both retrievals use look up tables (LUT) with LH profiles simulated by cloud-resolving models (CRMs) as function of precipitation rate and other parameters. However, an important difference between the two approaches is the source of these CRM simulations: for SLH, the LUTs are built from diabatic heating rates using CRM simulations with data from the Tropical Ocean Global Atmosphere-Coupled Ocean-Atmosphere Response Experiment (TOGA COARE) field campaign. Although TOGA COARE itself is an oceanic campaign,

the resulting SLH dataset is not restricted to oceanic regions, since the TRMM PR observes reflectivities over both land and ocean (Dorian, 2014). In contrast, the CSH retrieval uses a more diverse data in the CRM simulations to construct its LUTs, including data from six multi-week ocean field campaigns and four multi-week land field campaigns.

In addition to the differences in the data sources for constructing the LUTs of the LH profiles, the SLH and CSH algorithms also differ in their input variables, convective-stratiform classification methods, and how they handle stratiform rain:

In the SLH algorithm, the TRMM-PR input consists of precipitation top height (PTH), precipitation rate at the surface ($P_{srf}$) and the melting (freezing) level ($P_0$), as well as convective-stratiform classification. The convective-stratiform classification is based on the Goddard Cumulus Ensemble (GCE) method (Churchill and Houze, 1984), which identifies convective and stratiform regions using surface rain rate thresholds, cloud water content, and vertical velocity profiles from CRM simulations. The stratiform part is further separated into shallow stratiform and deep stratiform (anvil) based on the relationship between

PTH and the melting level height (4.4 km, Shige et al., 2004). The LUTs used in SLH are constructed differently for different rain types. For convective and shallow stratiform rain, the LUTs are based on PTH, while for deep stratiform rain, $P_0$ is used instead of PTH. In the latest SLH retrieval version (V6, Takayabu and Tao, 2020), deep stratiform rain is further divided into two subcategories: one where precipitation decreases from the melting level toward the surface, and one where precipitation increases. The first case represents typical stratiform rain where evaporation-driven cooling occurs below the melting level, and

the cooling magnitude is estimated from the difference between $P_0$ and $P_{srf}$. The second case, more common near convective areas, requires a separate set of LUTs, where the profile magnitude is determined directly by $P_{srf}$ (Tao et al., 2016; Takayabu and Tao, 2020).

In the CSH algorithm, surface precipitation rate, composite radar reflectivity, freezing level height, and echo-top height (ETH) are used as inputs. The CSH algorithm also applies a convective-stratiform classification, but the method differs from

SLH. In CSH V6 (Tao et al., 2022), the convective-stratiform separation follows a method more consistent with the GPM classification approach (Steiner et al., 1995). This method identifies convective and stratiform rain based on the radar reflectivity profile, comparing local reflectivity values to background averages, and detecting bright bands (sharp reflectivity decreases near the melting level). In CSH, stratiform profiles are further divided based on mean ETH, with deep stratiform defined as ETH > 5 km. To estimate low-level evaporative cooling in stratiform regions, CSH uses the vertical gradient of low-level reflectivity to distinguish between profiles with increasing and decreasing precipitation below the melting level (Lang and Tao, 2018; Tao

et al., 2022).

A comparison study by Tao et al. (2022) over five tropical (warm-season) regions shows that both retrievals capture the general structure of convective and the stratiform heating, with a broad heating in the middle troposphere in convective regions and heating aloft but cooling below in stratiform regions. However, SLH shows stronger heating aloft in the stratiform anvils

across all cases. In addition, the peak heating height in SLH profiles tends to be higher than in CSH, which is also reported by Elsaesser et al. (2022). This difference may be related to the fact that SLH is based on cloud-resolving simulations from the TOGA COARE field campaign, which represents oceanic convection with a larger fraction of stratiform rain (Tao et al., 2016). Another notable difference is that SLH-derived LH profiles show more structural details in stratiform regions, with three distinct heating peaks, while CSH-derived profiles generally show only two heating peaks over most regions.

In this investigation, we utilize the latent heating profiles of the GPM TRMM SLH dataset (V06), gridded at 0.5° latitude × 0.5° longitude and with a vertical resolution of 250 m (Shige et al., 2007). We used the unconditional LH profiles, averaged

over all measurements within each grid cell, and therefore averaging contributions from shallow and deep convection, from deep stratiform rain as well as from clear sky.

## 2.2 Cloud and atmospheric data

The AIRS instrument (Chahine et al., 2006) aboard the polar-orbiting satellite Aqua offers high spectral resolution measurements of the Earth's atmosphere at 1:30 AM and 1:30 PM LT since 2002. Its spectral coverage spans 2378 radiance channels within the wavelength range of 3.7–15.4 $\mu$m. AIRS footprints are grouped as $3 \times 3$ arrays. These arrays correspond to the size of the Advanced Microwave Sounder Unit (AMSU) footprints also on board. The spatial resolution of an AIRS footprint is about 13.5 km at nadir, and the swath width is approximately 1650 km. The latter leads to a substantial horizontal coverage of
approximately 70 % in the tropics at a specific local time.

The IASI instruments (Hilton et al., 2012) are operational on the European Metop platforms, starting data acquisition in 2007. They provide measurements at 9:30 AM and 9:30 PM LT. IASI is a hyperspectral and high-precision Fourier transform spectrometer. The 8461 spectral channels cover the infrared spectral domain from 3.6 to 15.5 $\mu$m. IASI footprints are grouped as $2 \times 2$ arrays. Again, these arrays correspond to the size of the AMSU footprints on board (August et al., 2012). The spatial
resolution of an IASI footprint is about 12 km at nadir, and the swath width is about 2200 km, leading to a 77 % coverage at a specific local time in the tropics.

The Clouds from IR Sounders (CIRS) retrieval reconstructs cloud properties from both AIRS and IASI measurements. It relies on a weighted $\chi^2$ methodology employing 8 channels in the vicinity of the 15 $\mu$m CO$_2$ absorption band as explained by Stubenrauch et al. (2010, 2017). The choice of 8 channels was made to establish a consistent long-term cloud climatology
by employing the same retrieval method across AIRS, IASI, and High-Resolution Infrared Radiation Sounder (HIRS) data (Stubenrauch et al., 2006). This retrieval simultaneously provides cloud emissivity ($\epsilon_{cld}$) and pressure ($P_{cld}$), along with associated uncertainties. CIRS cloud types are defined according to $P_{cld}$ and $\epsilon_{cld}$: high clouds ($P_{cld} < 440$ hPa), which are further categorized into high opaque clouds (Cb) with $\epsilon_{cld} > 0.95$, cirrus clouds (Ci) with $0.95 > \epsilon_{cld} > 0.5$, and thin cirrus (thin Ci) clouds with $0.5 > \epsilon_{cld} > 0.05$. UT clouds with $P_{cld} < 350$ hPa are part of the high cloud category. Mid-level clouds (440 hPa $<$
$P_{cld} < 680$ hPa) and low-level clouds ($P_{cld} > 680$ hPa) are further divided into opaque clouds with $\epsilon_{cld} > 0.5$ and partly cloudy with $\epsilon_{cld} < 0.5$.

For a consistent diurnal cloud variability from AIRS and IASI (Feofilov and Stubenrauch, 2019) the CIRS cloud retrieval uses auxiliary data (surface pressure and temperature, atmospheric temperature and humidity profiles, snow and sea ice information) from an identical source: ERA-Interim, obtained from the European Centre for Medium-Range Weather Forecasts (ECMWF)
meteorological reanalysis (Dee et al., 2011). These atmospheric profiles are also used to convert cloud pressure to cloud temperature ($T_{cld}$) and cloud height ($Z_{cld}$).

The ERA-Interim reanalyses have a spatial resolution of $0.75° \times 0.75°$ and are available at 00:00, 06:00, 12:00, and 18:00 UTC. They have been interpolated to the AIRS and IASI observation times by employing a cubic spline function. The spatial colocation was done in such a way that each array of footprints was associated with the closest ERA-Interim grid cell.

## 2.3 Input data for the artificial neural networks

Table 1 summarizes the input variables which are used in the artificial neural network predictions described in section 2.5.1.

Since the target data (TRMM-SLH) are given at a spatial resolution of $0.5°$, we adapt the input data by gridding them also to $0.5°$. The atmospheric properties (specific humidity and temperature profiles with a reduced vertical resolution containing 10 layers, total precipitable water, tropopause height) and surface properties (pressure and temperature) from the ERA-Interim ancillary data are averaged over $0.5°$, as well as cloud properties such as $P_{cld}$ and $\epsilon_{cld}$ and their uncertainties. In addition, we keep the cloud properties averaged over the most frequent scene (high-level or mid- and low-level clouds) in each grid cell. We also utilize the horizontal sub-grid structure within the grid cells: fractions of Cb, Ci, thin Ci, mid- / low-level clouds, and clear sky. The spectral variability of the effective cloud emissivity between 9 and 12 $\mu$m, computed by using the retrieved $P_{cld}$, indicates if the footprint is partly cloudy (Stubenrauch et al., 2017). Additional variables are the atmospheric window IR brightness temperature and its spatial variability, brightness temperature differences between atmospheric window and water vapour absorption channels, and the number of atmospheric layers down to surface.

Furthermore, we use additional variables from the CIRS-ML dataset (Stubenrauch et al., 2023) for evaluation and for scene identification, but they are not considered as input variables. As CIRS only identifies the uppermost cloud layer in the case of multiple layer clouds, the occurrence of a cloud underneath was deduced by a binary artificial neural network (ANN) classification per footprint, trained with CloudSat–CALIPSO layer information. A rain intensity classification (0 for no rain, 1 for light rain, 2 for heavy rain) was also obtained by ANN classification per footprint, but trained with precipitation rate data from CloudSat (2C-PRECIP-COLUMN, Haynes et al., 2009). The rain intensity classification considers light rain to be =< 5 mm h$^{-1}$ and heavy rain > 5 mm h$^{-1}$. In combination with a CloudSat 2C-PRECIP-COLUMN quality flag, which indicates certain rain, also expanded via a binary ANN classification, a 'rain rate indicator' was constructed and then averaged per $0.5°$ grid cell (Stubenrauch et al., 2023). The rain intensity classification used for the scene identification for the ANN training and production is based on this averaged rain indicator, with heavy rain starting probably already around 2.5 mm h$^{-1}$. We could show that this category corresponds to an average certain rain fraction of at least 0.8 (not shown), while the light rain category corresponds only to 0.3 over ocean (0.5 over land). The advantage of using the CIRS-ML rain intensity classification, is that the CIRS-ML data are available together with the whole CIRS-AIRS and CIRS-IASI data records, so we can use it for a scene identification for the training as well as for the application of the ANN models developed for these different scenes discussed in section 2.5.1.

**Table 1.** List of input variables for the prediction of latent heating: gridded over $0.5° \times 0.5°$

| Input variables | Definitions |
|---|---|
| **Clouds** | |
| $\epsilon_{\text{cld}}, P_{\text{cld}}, T_{\text{cld}}, Z_{\text{cld}}, d\epsilon_{\text{cld}}, dP_{\text{cld}}$ | CIRS cloud properties and uncertainties |
| $\epsilon(\text{scene}), P(\text{scene}), \text{frac\_scene}$ | CIRS cloud properties and fraction of most frequent scene |
| $\sigma(\epsilon(\lambda_i)))$ | Spectral variability of effective cloud emissivity over six wavelengths (9-12 $\mu$m) |
| **Atmosphere** | |
| $\text{TB}(11.85\mu m), \sigma(\text{TB})$ | AIRS / IASI brightness temperatures and spatial variability |
| $\text{TB}(11.85\mu m) - \text{TB}(7.18\mu m), \text{TB}(11.85\mu m) - \text{TB}(7.23\mu m)$ | Brightness temperature difference between atmospheric window and water vapour absorption channels |
| $q, T$ within 10 layers | ERA-Interim specific humidity and temperature profile |
| total precipitable water, $P_{\text{tropopause}}$ | ERA-Interim column water vapour, tropopause height |
| **Surface** | |
| $P_{\text{surf}}, T_{\text{surf}}, \text{nb of atm. layers down to } P_{\text{surf}}$ | ERA-Interim surface properties |
| **Horizontal sub-grid structure of CIRS cloud types from footprints** | |
| $\text{frac\_Cb}, \text{frac\_Ci}, \text{frac\_thCi}$ | Fraction of cumulonimbus, cirrus, thin cirrus |
| frac\_mlow | Fraction of mid and low level clouds |
| frac\_clr | Fraction of clear sky |

*scene: 1 for high level clouds and 2 for mid-low level clouds.

## 2.4 Collocation of input and target data

Due to their different orbit characteristics, the observations from the TRMM, Aqua and Metop satellites seldom coincide in both time and geographical location when observing the Earth. Consequently, only a subset of their respective observational databases can be overlaid and employed for the training. We allow a 20-minute time differential between the CIRS and TRMM data. AIRS orbits (1:30 AM and 1:30 PM) and IASI orbits (9:30 AM and 9:30 PM LT) are independently co-located with TRMM orbits. Figure 1 illustrates the different swath width of the Aqua–AIRS and TRMM–PR measurements and their coincidence for a specific day at a specific observation time. Within a time window of $\pm 20$ minutes, the overlap between the TRMM-PR and the AIRS measurements is only 3 %. Our goal is to expand this coverage to about 70 % and 77 % (swath coverage of AIRS and IASI, respectively) by applying ANN methods to the input data described in section 2.3. This allows us then to relate horizontal fields of clouds, LH and $Q_{rad}$ at specific local times, as shown in section 3, as well as the MCS analysis described in section 5.2.

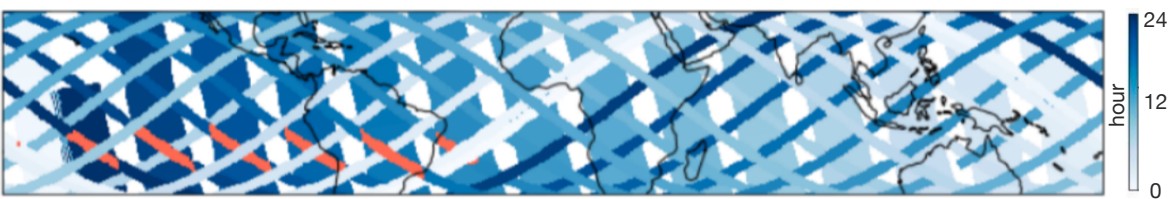

**Figure 1.** Illustration of the temporal and spatial match between Aqua–AIRS and TRMM–PR orbits. Orange represents the satellite trace segments, coincident within a time window of $\pm$ 20 minutes, for one specific day. The narrow swaths represent TRMM–PR orbits, while the broader swaths represent Aqua–AIRS orbits. Shades of blue indicate variations in sampling time difference.

The co-located AIRS–TRMM data spanning from 2004 to 2013 consist of approximately 2,300,000 cases, and the IASI–TRMM data from 2008 to 2014 contain about 1,600,000 cases, for each of the AM/PM measurement times.

## 2.5 Artificial Neural Network predictions and evaluation

### 2.5.1 Development of prediction models

As the distribution of precipitation rates is very skewed, with a large peak at 0 mm h$^{-1}$ and a very long tail towards larger values, we first examine the shapes and statistics of the TRMM–SLH latent heating profiles within the collocated data.

In general, convective towers which produce a strong latent heating can be identified by a heavy rain rate. However, these occur much less frequently (10 %) than scenes of lightly precipitating (25 %) and non-precipitating (65 %) high-level clouds. Over land the diurnal variation of these convective towers is larger than over ocean, as expected, with a minimum in the morning (7 %) and a maximum in the evening (12 %), while non-precipitating high-level clouds vary from 65 % at night to 77 % in the afternoon. The distinction between non-precipitating, lightly and heavily precipitating is given by the rain intensity classification described in section 2.3. For the training of the artificial neural network regression models, the least frequent scenes, the ones with heavy precipitation (see above), will be less represented, but these are the scenes we are most interested in (for the MCS studies in section 5).

Figure 2 presents LH profiles of high-level clouds, separately averaged over these three rain intensity scenes and over ocean and land, at the four CIRS observation times. Though the rain intensity categorization is derived from CloudSat radar and ANN classification using CIRS and ERA-Interim data, the shape of the TRMM LH profiles are consistent with this scene classification, for both ocean and land cases. As expected, the LH profiles of these three scene types differ considerably, with nearly no LH for no rain and very large LH for heavy rain. The latter show large peaks in LH around 450 hPa, with a larger maximum over ocean (30 K day$^{-1}$) than over land (approximately 20 K day$^{-1}$). This may be linked to smaller systems over land than over ocean (e.g., Liu, 2007). A larger diurnal variation over land is also observed, as expected.

When considering all precipitating high-level clouds or even all precipitating clouds (Fig. S1), the averages are far smaller than the ones of heavy precipitation, because the latter occurs only rarely. Since the LH profile shapes and statistics of the different rain intensity categories are very different, we developed ANN models for each of the three classes in rain intensity,

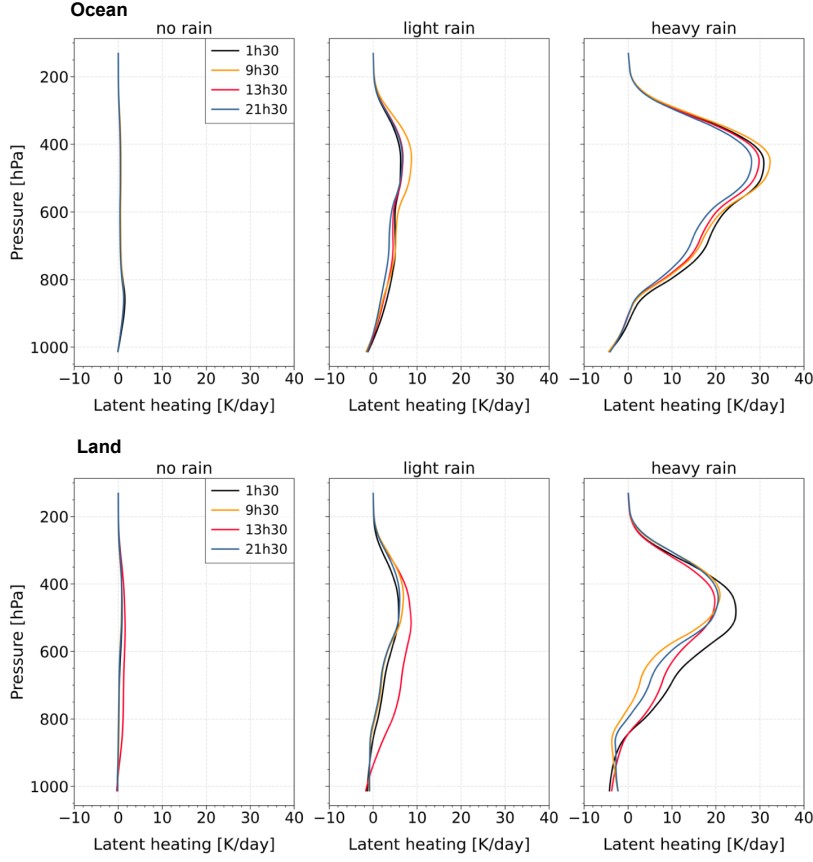

**Figure 2.** TRMM-SLH LH profiles of high-level clouds categorized by CIRS-ML rain intensity as described in section 2.3, at 4 observation times (1h30, 9h30 AM/PM), over ocean (top panels) and over land (bottom panels), averaged over the TRMM–CIRS collocated data during the period 2008–2013, within 30°N–30°S, at a spatial resolution of 0.5°.

separately over ocean and over land. If we would have trained only one single ANN regression model of all scenes together, the LH of heavy rain would have been underestimated because of the small statistics (as has been shown for radiative heating rates in the case of all clouds and Cb alone in Stubenrauch et al. (2021). For each of these precipitation classes, we also separated

scenes of high-level clouds and of mid- and low-level clouds. The latter show a peak of only about 5 K/day at about 900 hPa in the case of heavy rain (Fig. 5b of Stubenrauch et al. (2023), Fig. S6 and S8). This leads us to develop 12 ANN regression models to predict LH profiles from AIRS or from IASI, respectively. The predicted LH rates are given at a spatial resolution of 0.5° and at a reduced vertical resolution of 20 layers, given within the same pressure layers as the CIRS-ML radiative heating rates (Stubenrauch et al., 2021).

The final ANNs of the regression models consist of 2 hidden layers both with 32 neurons and one output layer, as shown in Fig. S2. We applied the Rectified Linear Unit (ReLU) activation function on the hidden layers. In addition, we used the

Root Mean Square Propagation (RMSprop) optimizer with a learning rate of 0.0001 and a batch size of 512. Furthermore, we applied a Min-Max Normalization to the variables. The collocated dataset was randomly divided into three categories: 60 % for training, 20 % for validation, and 20 % for test. The latter two have not been directly used in the training: While the validation data are used in the iteration process of the training, the test data are used for evaluation after the training (section 2.5.2).

We use the Mean Squared Error (MSE) as a loss function for the regression models, which evaluates the mean squared difference between observed and predicted values. The model parameters are fitted by minimizing this loss function. The Mean Absolute Error (MAE) is the average of absolute differences between observed and predicted values, and it is used to assess the quality of these regression models. In order to avoid overfitting, we stop training when the loss function does not improve for 20 iterations. We compare the behaviour of the loss function of the training data with the one of the validation data. Similar results suggest that the model performs consistently well on both datasets, showing a certain level of generalization ability. Figure S4 shows a good performance, except for a slight overfitting for heavy precipitation.

The hyper-parameter selection of the neural network may influence the training results. Therefore, we tested different deep learning parameters (e.g., number of neurons per layer, number of layers, learning rate, regression kernel) to optimize these ANN regression models. The results show only slight differences in training/validation loss and MAEs among these tests, and there was no significant impact on the LH results.

To study the impact of the variable selection on the predictions, we have experimented with various input variables, in particular we included and excluded information on the rain intensity substructure within the grid boxes of $0.5°$. By using this information, the results were slightly better, in particular the spread of the predicted LH was slightly larger. However, this approach resulted in a slight positive anomaly in vertically integrated LH for March 2014 and later on, when considering the time series of LH derived from AIRS data (not shown). The LH time series derived from IASI were not affected. In March 2014, the AIRS instrument suffered from a solar flare event which led to tiny artifacts which show in the ANN rain rate classification (Stubenrauch et al., 2023). Consequently, we have excluded these grid fractions of different rain intensity classes from our input variables.

We also assessed the effectiveness of the fraction of clouds below CIRS clouds. For non-precipitating high-level clouds in Fig. 2, the very small LH peak of less than 1 K day$^{-1}$ at about 900 hPa can be explained by lower clouds underneath which produce rain (Fig. S3). Over land, this lower small peak is vertically more extended and only observed at 13:30 LT. This may be due to developing convection, which has its peak in the late afternoon, and is missed by the AIRS and IASI observation times (see section 3). While this CIRS-ML variable effectively classifies low clouds underneath (Fig. S3), it did not improve the results as additional input.

Therefore, we kept the configuration which uses the 27 variables given in Table 1 as input variables.

## 2.5.2 Evaluation using test data

We have evaluated the LH prediction results by comparing them with those of the target TRMM-SLH, using the 20 % test data of the collocated data.

For both AIRS and IASI, the MAE values, presented in Table S1, are notably small, varying from 0.02 K day$^{-1}$ for non-precipitating to 0.55 for heavily precipitating high-level clouds. Moreover, the loss function, presented in Fig. S4, decreases rapidly with the iterations (epochs), followed by stabilization.

As shown in the upper panels of Fig. 3 and S6, as well as Fig. S5 and S7, the predicted LH profiles capture on average exactly the specific patterns of the high-level clouds:

(1) For the heavy rain case, high-level clouds produce a noticeable 25 K day$^{-1}$ LH at about 450 hPa.

(2) In the light rain case, the predicted LH exhibits a much flatter distribution with only a very small peak, of about 5 K day$^{-1}$ at around 450 hPa.

(3) In the no-rain case, LH is close to 0 K day$^{-1}$, with a tiny peak of approximately 0.7 K day$^{-1}$ in the low-altitude region (P > 800 hPa), corresponding to low-level clouds, mostly underneath non-precipitating cirrus, as suggested by Fig. S3, which

shows a twice as large peak when only selecting cases with clouds below according to a classification from CIRS–ML (see section 2.5.1).

While the averages of the predicted results of the different scenes agree well with those of the target TRMM-SLH data, the spread of the predicted values is much smaller. The TRMM-SLH data reveal a large variability between 550 and 900 hPa, which may be linked to the variability between stratiform and convective rain within the 0.5°, as well as due to additional

precipitation from underlying clouds. This loss in the prediction of the variability may be due to insufficient input information and a too coarse spatial resolution. but also because of the skewness of the precipitation distribution itself.

Compared to the LH prediction, the lower panels of Fig. 3 show the average CIRS-ML radiative longwave (LW) heating profiles and the target CloudSat–CALIPSO FLXHR profiles (Henderson et al., 2013) as well as their spread, separately for thin cirrus, cirrus and high opaque clouds. Here the spread of the predicted and target data is similar, in particular for the

optically thicker clouds. Relatively opaque clouds contribute to heating the atmospheric column below by trapping surface emissions while inducing cooling effects in the column above due to excess emission. In contrast, thin cirrus clouds warm the UT by intercepting LW radiation originating from below (Stubenrauch et al., 2021). Notably, above Cb clouds, the cooling is on average -4.5 K day$^{-1}$ around 170 hPa. The small cooling observed at approximately 550 hPa is attributed to the melting process, which takes place at or just below the freezing level, typically around 5 km above sea level across tropical regions.

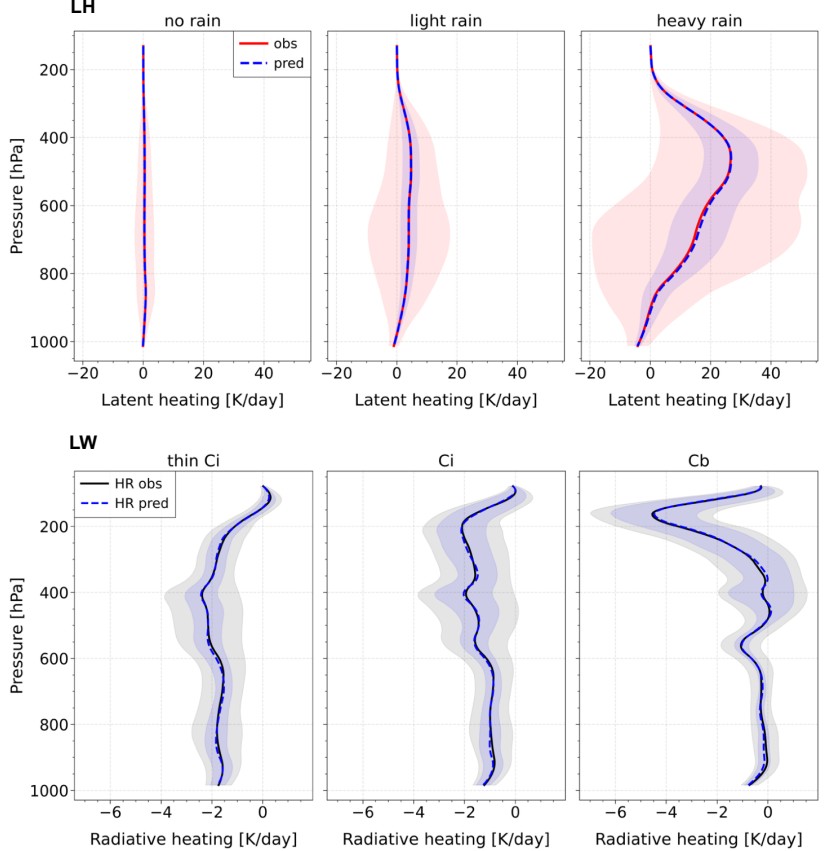

**Figure 3.** Average and range of predicted (dashed) and target (solid) latent heating (2004–2013) and radiative heating (2007–2010) of high-level clouds, within 30°N–30°S, at a spatial resolution of 0.5°. ANN models were trained on collocated AIRS–TRMM and AIRS–CloudSat–CALIPSO data, respectively. Upper panels: Latent heating rates for different rain rate intensities (no rain, light rain and heavy rain); lower panels: radiative longwave heating rates for different UT cloud types (thin cirrus, cirrus and cumulonimbus). Shaded areas indicate ±67.5 % of the standard deviation, which approximately corresponds to quartiles.

Figure S7 shows that the LH profiles predicted from AIRS have slightly larger values in the lower troposphere (below 500 hPa) than the ones predicted from IASI, in particular over land. This may be explained the diurnal cycle of precipitation coming from clouds at different height, since both AIRS and IASI predictions are on average in excellent agreement with the target TRMM-SLH profiles.

## 3 Construction of the 3D Latent heating dataset by applying the ANN models

During the training process, ANN models are tuned to fit the data through continuous adjustment of hyper-parameters by the optimization algorithm to minimize the difference between the predicted output and the target data (loss function). We use these trained models to extend the TRMM–SLH data, which only cover about 3 % of the CIRS observations, to a coverage of 70 % and 77 %, corresponding to the AIRS and IASI swaths over the period 2004–2018 and 2008–2018, respectively. We evaluate the effect of using only specific local times (given by the ANN-predicted LH at observation times of AIRS and IASI), and we compare the vertically integrated LH (LP) to the original temporal sampling of TRMM. At the end of this section, we illustrate how the ML-derived data can be used to compare the horizontal and vertical structure of LH and $Q_{rad}$ between La Niña and El Niño.

### 3.1 Coherence in diurnal variation

Figure 4 shows the vertical LH profiles from the ML production for precipitating clouds. We consider all clouds and specifically high-level clouds, separately over ocean and over land. AIRS and IASI observations allow us to examine these profiles at four distinct observation times. The results are coherent with expectations on the diurnal variation:

(1) The shape of the LH profiles averaged over precipitating clouds strongly differs between ocean and land: Over ocean, the mean LH profiles have two peaks, at 450 hPa and at 850 hPa, corresponding to the contributions from high-level clouds and low-level clouds (Shige et al., 2004), while over land the LH is mostly produced by high-level clouds.

(2) The diurnal spread is larger over land than over ocean, as expected. Over land, the precipitation frequency typically peaks in the late afternoon, whereas over ocean, the diurnal cycle is less pronounced with a maximum occurring in the early morning (e.g., Nesbitt et al., 2000; Dorian, 2014). The observations at 1:30 PM are before the peak of land precipitation, while those at 1:30 AM are before the early morning peak of ocean precipitation. LH also involves the intensity of precipitation, and given that MCSs contribute to over half of intense precipitation (e.g., Roca et al., 2014), we observe the largest peaks at 450 hPa (corresponding to stratiform anvil precipitation) at 9h30 PM over land and at 9:30 AM over ocean, respectively, a few hours after convection had started and MCSs could form.

(3) In particular over land, we notice a diurnally changing profile shape, with a stronger contribution from lower atmospheric levels at early afternoon, corresponding to the development of cumulo-congestus. Later in the evening, the peak moves higher into the UT, corresponding to stratiform anvils. During night there may be a more complex vertical structure with convection from lower clouds underneath the high-level clouds, and LH is minimum in the morning.

Small differences with the original TRMM-SLH averages in Fig. S1 may be explained by different spatial sampling (70 / 77 % coverage compared to 3 % coverage).

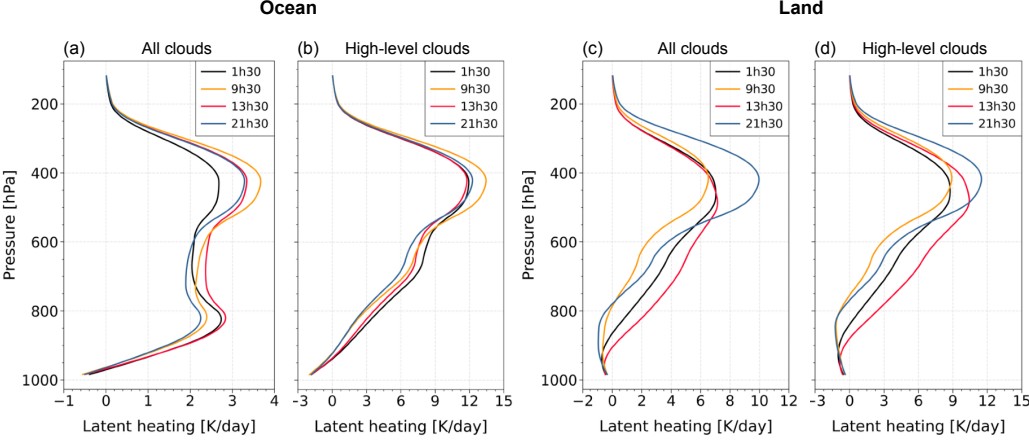

**Figure 4.** Average CIRS-ML LH profiles of all precipitating cloud scenes (a,c) and precipitating high-level clouds (b,d) at 4 observation times (1h30, 9h30 AM/PM), separately over ocean (a,b) and over land (c,d). Data are from the CIRS–ML production during the period 2008–2013, within $30°$N–$30°$S, at a spatial resolution of $0.5°$.

## 3.2 Coherence between TRMM-SLH LP and ML-predicted LP

Due to the large diurnal cycle of occurrence and intensity of precipitation over land, the time sampling plays an important role in order to obtain reliable monthly means of LH. Figure 5 presents zonal averages of the vertically integrated LH (LP): ML-derived LP at specific observation times and original TRMM-SLH LP with its specific diurnal sampling, separately over ocean and over land.

We note the following points:

(1) Over ocean, all zonal means of LP show a peak around $5°$ north of the equator, with a small broader peak from the equator to approximately $10°$S, corresponding to the large LP released over the Pacific warm pool. Over land, there is only one broad peak from about $10°$N to $10°$S.

(2) Overall, the latitudinal behaviour of LP given by AIRS–ML and IASI–ML is consistent with the one given by TRMM-SLH with a broader diurnal sampling. It is remarkable that over ocean the zonal averages of LP at 1:30 AM and 1:30 PM agree very well with those from TRMM-SLH. However, over land, as expected, considering LP only at these two observation times underestimates the daily mean LP, because the strong convection in the afternoon is not captured. The effect is the worst at 9:30 AM.

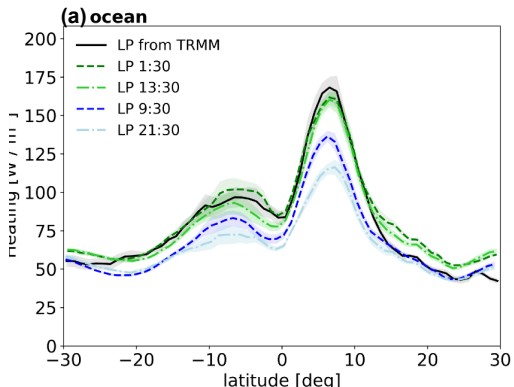 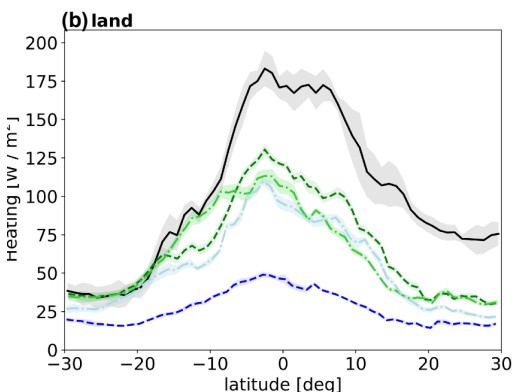

**Figure 5.** Zonal averages of vertically integrated LH (LP) at 4 specific observation times (1h30, 9h30 AM/PM) from the CIRS-ML production and for the original TRMM, including diurnal sampling, (a) over ocean and (b) over land. Latitude intervals are 1°. LP from TRMM-SLH data for the period 2008–2013 is represented by a solid black line. The green dashed line and bright green dash-dotted line represent LP from ML regression using TRMM–AIRS (2008–2013) as inputs, at 1:30 AM and 1:30 PM, respectively. The blue dashed line and light blue dash-dotted line represent LP from ML regression using TRMM–IASI (2008–2013) as inputs, at 9:30 AM and 9:30 PM, respectively. Shaded areas correspond indicate inter-annual variabilities.

Furthermore, Fig. S9 compares the LP zonal means of the ML-derived data with the collocated TRMM-SLH data, separately over ocean and over land, at the four observation times. One has to keep in mind that the spatial sampling is much smaller than the one in Fig. 5. In general, the LP zonal averages from ML agree quite well with that from the collocated data, with small explainable biases: slight underestimation in regions with strong rain (tropical peak region) and very slightly overestimation in regions with not much rain (subtropics).

To investigate the coherence of LP monthly averages over grid cells between TRMM–SLH and CIRS–ML in more detail, we examined their relationship at different scales over ocean, using averages over 1°, 2.5°, 5° and 10°. Differences from a 1-to-1 relationship stem (1) from biases in the CIRS–ML LH and (2) from differences in the sampling of observation times. Since the diurnal sampling of TRMM is not homogeneous, a larger grid cell has a larger probability to include more observations at 1:30 LT. As the TRMM revisit cycle depends strongly on latitude Negri et al. (2002), with 23 days at the equator and up to 46 days at the highest latitudes (the latter should have different observation times sampled in different months), we limited the latitudinal band to 10°N–10°S for this comparison. We computed the slopes and correlation coefficients between monthly mean LP of TRMM-SLH and CIRS–ML, averaged over different scales, and Fig. 6 presents the normalized density in the CIRS-ML LP and TRMM-SLH LP plane. While the slopes increase from 0.54 to 0.82 in this latitude band, they vary from 0.44 to 0.77 at the higher latitudes (20°-30° N and S, not shown), where the TRMM repeat cycle is only half. The larger slopes in the latitude band nearer to the equator demonstrate their dependency on the TRMM diurnal sampling variability. The increase of the slopes and the increasing linearity of the points with increasing grid cell size in the CIRS-ML LP and TRMM-SLH LP show a strong bias and noise reduction when averaging over more observations within a grid cell. At the spatial averaging over 1°, the small

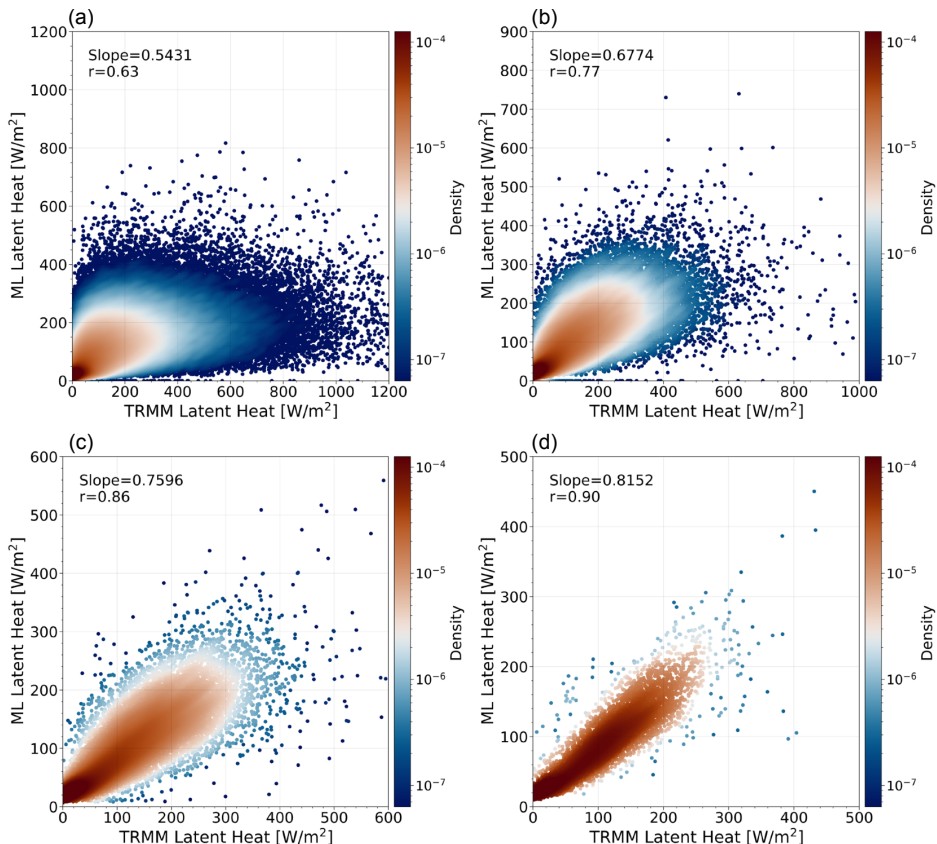

**Figure 6.** Relationship of LP monthly means per grid cell between TRMM–SLH LP and of CIRS–ML at different spatial scales: (a) $1°$ latitude × $1°$ longitude, (b) $2.5° × 2.5°$, (c) $5° × 5°$ and (d) $10° \times 10°$. Statistics over ocean $10°$N–$10°$S, over the time period from 2004–2013.

slope is related to the fact that the ANN regression does not capture well extreme events, mainly because of insufficient input information and the skewed distribution of input data, as mentioned in section 2.5, and the relatively noisy relationship is due to the inhomogeneous TRMM sampling at specific times. Therefore, for larger grid cell sizes the agreement is much better.

In summary, the increasing slopes of the relationship between LP of CIRS–ML and TRMM-SLH suggest that our ML-expanded LH dataset is suitable at scales larger than about $2.5°$ (with a slope of 0.68). At a scale of $1°$, one observes still a correlation, but the relationship is much noisier and CIRS-ML LP is strongly underestimated for large TRMM-SLH LP. This can be explained by the fact that extreme values are very rare and as seen in Fig. 3 the ANN regression is capable to produce a reliable mean per scene, but the training statistics of precipitation extremes most probably was too small.

In order to further understand the noise and biases, we analysed the complete statistics of the collocated data in order to link discrepancies in LP with those in rain fraction over the $0.5°$ grid cells estimated from CIRS-ML (CloudSat) and TRMM. Therefore, we compare CIRS-ML certain rain fraction as function of TRMM rain fraction and TRMM-SLH LP (Fig. S10). Coherently, the average certain rain fraction increases with both, as one would expect. When separating cases with small and

large fraction of certain rain within the grid cells, the CIRS-ML LP is much larger for a large fraction, as one would again expect. Cases with a large certain rain fraction for very small TRMM-SLH LP are rare, but their occurrence leads to a slight overestimation. The distributions of CIRS-ML certain rain fraction in the CIRS-ML LP–TRMM-SLH LP space, shown in Fig. S11, explain then the noise, in particular for small TRMM LP. So, this noise can be mostly explained by a few individual cases which show a mismatch between the certain rain coverage obtained from ANN classification of certain rain identified over CloudSat samples (1.25 km × 2.5 km), and the TRMM radar samples (5 km × 5 km). Nevertheless, what is important to note is that the CIRS-ML LP seems to be coherent with the certain rain coverage, even though this variable was not used in the training.

## 3.3 Structure of diabatic heating: Contrasting La Niña and El Niño events

After verifying the ML-derived LH production along the vertical and latitudinal directions, we illustrate that the horizontal patterns produced by this 3D LH dataset for La Nina and El Nino are as expected. Figure 7a–d shows geographical maps of LP and ACRE respectively, for two distinct scenarios: La Niña (JAN 2008) and El Niño (JAN 2016). LP is marked by contours while the colours correspond to ACRE values. The LH and CRE profiles, averaged over 30°N–30°S, are given as a function of longitude in Fig. 7e and 7f.

During El Niño (warm phase), sea surface temperatures (SST) are higher than normal in the eastern and central equatorial regions of the Pacific Ocean, leading to increased convection and cloudiness in these regions, which cause an increase in latent heat release. Lower SST in the western Pacific leads to a decrease in convection and cloudiness in the region, resulting in negative LP anomalies. La Niña shows the opposite behaviour (Fig. 7a and 7b).

During La Niña, there is a large, structured band of latent heating in the South Pacific Convergence Zone (SPCZ) (Fig. 7c and 7d), in addition to large LP in the Intertropical Convergence Zone (ITCZ), mostly over the continents. During El Niño, we notice that the maximum distribution of latent heating moves eastward, which happens because the upward branch of the Walker Circulation shifts towards the central Pacific (e.g., Bayr et al., 2018). This is even more evident in Fig. 7e and 7f. Furthermore, there are some regions with negative CRE values at altitudes around 850 hPa, which correspond to the presence of low-level clouds. This effect is more significant during La Niña because convective activity peaks in the western Pacific during La Niña, whereas in the eastern Pacific, with lower SSTs and less convective activity, there is more low-level cloud formation, thereby leading to a negative CRE aloft.

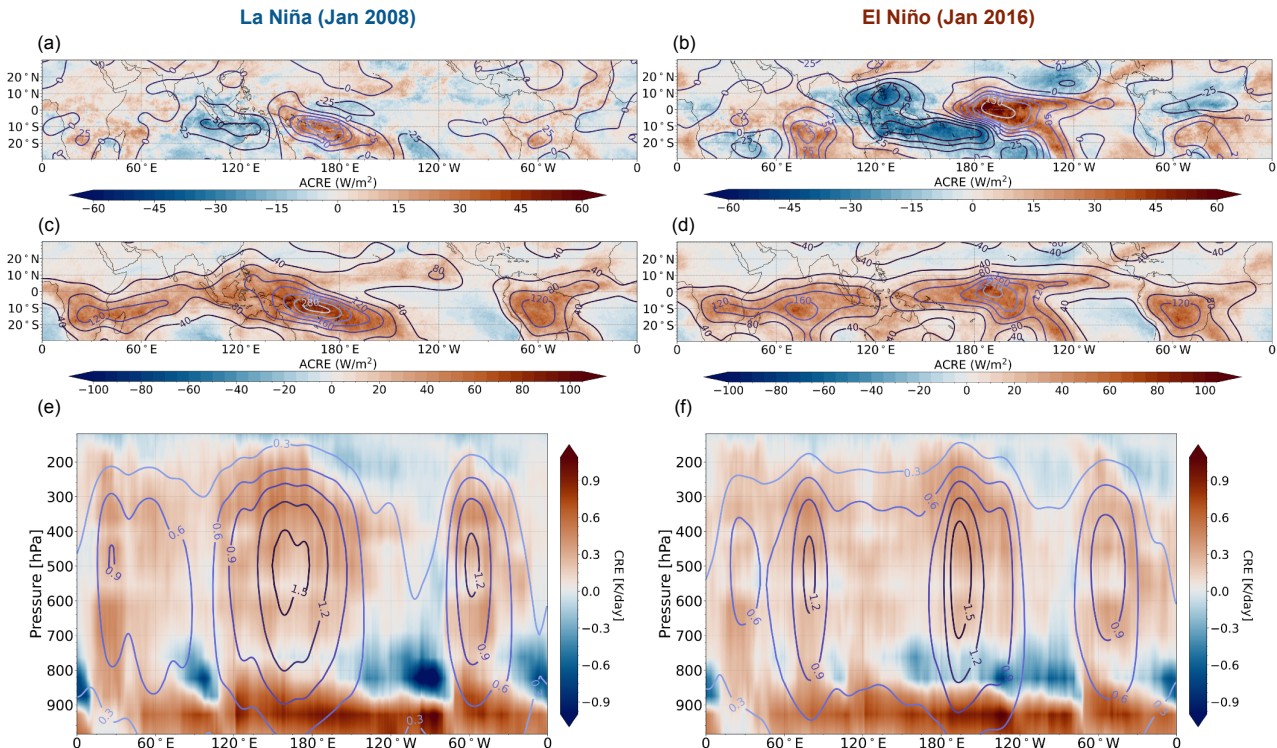

**Figure 7.** (a) & (b): Maps of anomalies of vertically integrated LH and ACRE during La Niña (JAN 2008) and El Niño (JAN 2016), obtained by subtracting the corresponding 11-year (2008–2018 JAN) averages; (c) & (d): Maps of vertically integrated LH and ACRE during La Niña (JAN 2008) and El Niño (JAN 2016), respectively; (e) & (f): Plots of longitudinal-vertical LH and CRE during La Niña (JAN 2008) and El Niño (JAN 2016), respectively. All data are within 30°N–30°S, at a spatial resolution of 0.5°. The colorbars represent the ENSO ACRE anomalies (W m$^{-2}$), ACRE (W m$^{-2}$) and CRE (K day$^{-1}$) values from CIRS–ML–Cloudsat–CALIPSO, while contours correspond to vertically integrated ENSO LH anomalies (W m$^{-2}$), vertically integrated LH (W m$^{-2}$) and LH values (K day$^{-1}$) from CIRS–ML–TRMM, at 1h30 AM/PM (AIRS).

We also assessed the geographical coherence between the CIRS-ML predicted LP and the LP calculated using precipitation data from TRMM (3B42_Daily, Huffman et al., 2007) and the Global Precipitation Climatology Project (GPCP CDR_V2.3,
Adler et al., 2016) during the same ENSO events. To facilitate comparison between our ML-derived LP and the precipitation datasets, we estimated $LP = \rho_l L\nu R$, as in L'Ecuyer and Stephens (2007): where $\rho_l$ is the density of liquid water (1000 kg m$^{-3}$), $L\nu$ is the latent heat of vaporization of water ($2.5 \times 10^6$ J kg$^{-1}$), and $R$ represents the surface rainfall rate (m s$^{-1}$). Using this formula, the conversion factor to convert precipitation rates from TRMM and GPCP (mm day$^{-1}$) to latent heat flux LP (W m$^{-2}$) is 28.9 W m$^{-2}$/(mm day$^{-1}$). Figure 8 shows that the geographical patterns as well as the absolute values of the
monthly mean LP computed from GPCP and from TRMM daily accumulated precipitation agree very well. Moreover, the geographical patterns of our ML-predicted LP align closely with the ones derived from TRMM and GPCP precipitation data during both La Niña and El Niño phases, though the absolute values in the regions of larger LP are underestimated as shown

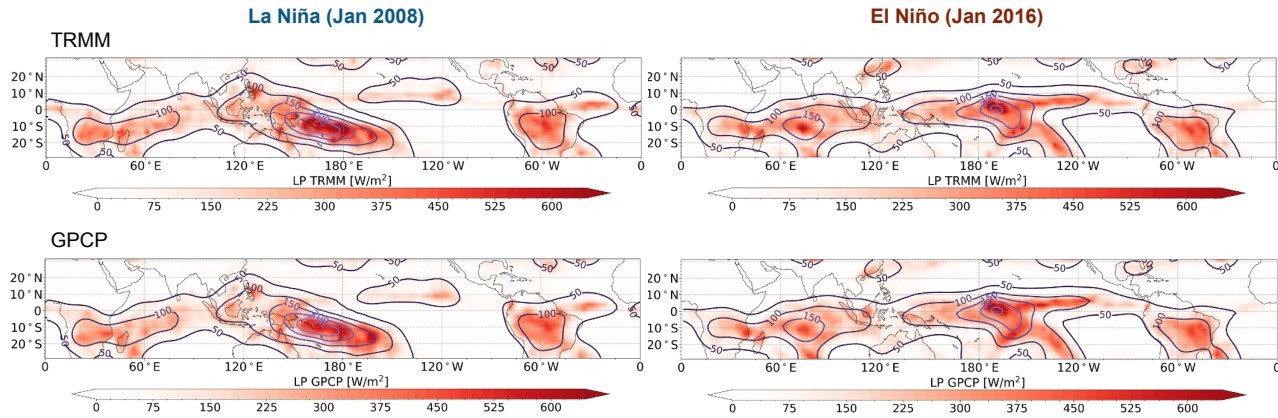

**Figure 8.** Comparison of horizontal structures between the vertically integrated LH (LP) of CIRS–ML (at 1:30 LT) and those obtained from daily precipitation accumulation of TRMM (upper panels) and GPCP (lower panels) during La Niña (JAN 2008) and El Niño (JAN 2016), within the region of 30°N–30°S, at a spatial resolution of $2.5° \times 2.5°$. The LP from CIRS–ML is represented by contours, while the colorbars indicate the LP from TRMM and GPCP.

by Fig. S13, presenting the ratio between CIRS-ML LP and TRMM (or GPCP) LP for grid cells with a TRMM (or GPCP) LP larger than 50 W m$^{-2}$. This was expected already from the results presented in section 3.2. Nevertheless, the CIRS-ML LH allows to study monthly mean spatial patterns at four specific times, keeping in mind that the range between smallest and largest LP is underestimated. This is not possible with the original TRMM dataset, which does not fully cover the latitudinal band over a whole month at a specific time, as demonstrated in Fig. S12.

All the above suggests that the monthly means of the CIRS-ML-expanded LP well represent the horizontal structures seen in other datasets and strongly reflect the characteristics of ENSO events. In addition, ACRE shows a highly matched distribution pattern with LP. In other words, larger ACRE distributions are also seen in the regions of larger LP, and therefore enhancing LP. In section 5 we will explore in more detail the connection between them, in particular at the scale of mesoscale convective systems (MCS). For this study, we need to average LP and ACRE over the horizontal extent of the MCSs. Therefore, we describe in the next section the reconstruction of these MCSs.

## 4 Construction of mesoscale convective systems

The study in section 5.2 needs the identification of mesoscale convective systems, including their non-precipitating anvil parts as these also provide radiative heating. Therefore, we use the CIRS data to reconstruct first UT cloud systems, with a method developed by Protopapadaki et al. (2017) and refined by Stubenrauch et al. (2023). We consider UT clouds with $P_{cld} < 350$ hPa. The grid cells of $0.5°$ latitude $\times 0.5°$ longitude have to be covered by at least 90 % of these clouds. Since ubiquitous thin cirrus in the TTL (Tropical Tropopause Layer) connect with many of the MCS, we exclude for this UT cloud system

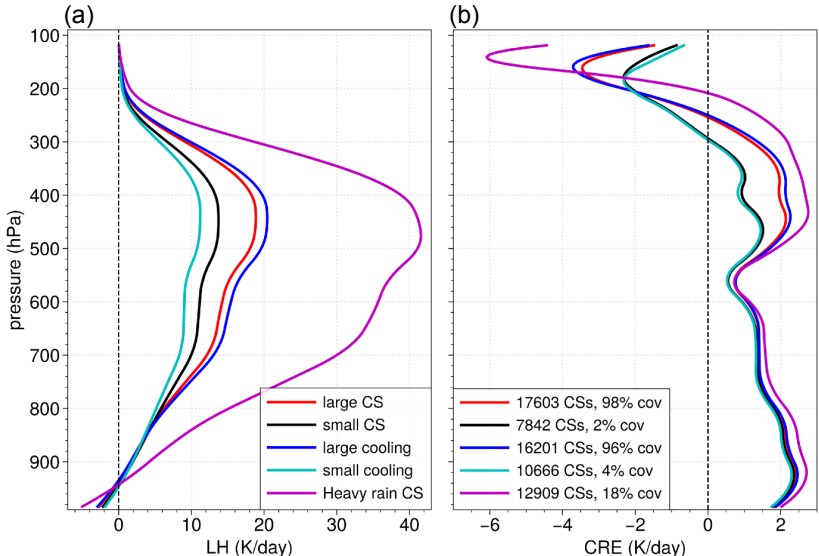

**Figure 9.** (a) Profiles of latent heating; (b) profiles of CRE. Averages over CSs with the presence of at least one grid cell with heavy precipitation (magenta), averages over small CSs (black) and MCSs (red), and averages over CSs with a small (cyan) and large (blue) LW cooling above the precipitating parts. The classification thresholds are 12,000 km$^2$ for size and -7.5 K day$^{-1}$ for the minimum of the LW cooling above the precipitating part of the CS, respectively. The number and their coverage compared to all CSs are also given per category. Oceanic systems over the time period 2004–2018, within 25°N–25°S.

reconstruction those UT clouds with emissivity smaller than 0.2. First, adjacent grid cells with UT clouds of similar height (within 8 hPa × ln($P_{cld}$ [hPa])) are merged together into the same system. Then the size of convective cores is determined first by counting the number of grid cells with cloud emissivity > 0.98 within regions where the cloud emissivity exceeds 0.93 and then by multiplying this number by the grid cell size of 0.5° (approximately 3000 km$^2$). We define a convective system as an UT cloud system with at least one convective core and the presence of precipitation. Earlier studies (Protopapadaki et al., 2017;

Stubenrauch et al., 2019, 2023) have shown that the convective core fraction, given by the ratio of convective core size over MCS size, can be used as a proxy of the maturity stage.

Since for the cloud system reconstruction the gaps between the orbits have been filled (Protopapadaki et al., 2017), but the CIRS–ML diabatic heating and precipitation intensity classification have only been extended within the orbits of AIRS and IASI, we select for the following analyses only systems for which there is an overlap with the ACRE and LP swaths by

more than 70 %. Furthermore, we concentrate only on oceanic systems, defined as systems with less than 10 % of their size overlapping land, and we limit the latitude band to 25°N-25°S, because most of these systems are located there according to Fig. 5 of Protopapadaki et al. (2017). These criteria leave us with about 26358 convective systems (CSs) for the period of 2004–2018.

In order to test the coherence of the data, we compare the diabatic profiles of different categories of convective systems.

Therefore, we explore various proxies of convective intensity: (1) the presence of heavy precipitation (e.g., Takahashi et al.,

2021), (2) MCS size (e.g., Roca and Fiolleau, 2020; Stubenrauch et al., 2023, etc) and (3) the minimum of the LW cooling above the precipitating part of the MCS. The latter is directly linked to the opacity: the larger the LW cooling above the cloud the denser the cloud itself (Stubenrauch et al., 2021).

Figure 9 presents profiles of latent heating and CRE averaged over CSs of different precipitation intensity distinguished by the proxies described in the paragraph above and using the thresholds given in the figure caption. All profiles show a peak in latent heating at around 450 hPa. This is much higher than one would expect from an isolated convective tower, and is linked to additional stratiform rain of the thick anvils (i.e., Hartmann et al., 1984; Schumacher et al., 2004; Chang and L'Ecuyer, 2019). The CSs including heavy precipitation, covering about 20 % of the area of all CSs, produce the largest LH, with a maximum of about 40 K day$^{-1}$ around 450 hPa and a broad shoulder downwards to 700 hPa. CSs with a large size (MCSs) or with a strong LW cooling above their precipitating parts also show a larger LH than those of smaller size or with a smaller LW cooling. Radiative heating adds a small positive heating from 200 hPa downward and a cooling above the opaque parts of the CSs. Cooling and heating are much stronger for CSs including heavy precipitation, leading to strong vertical gradients.

The size threshold of 12000 km$^2$ corresponds to approximately four grid cells of 0.5°, and it distinguishes between CSs and MCSs, the latter with a statistics of 17603 and covering about 98 % of all CSs identified by CIRS.

## 5   Reinforcement of latent heating by UT cloud radiative heating

Over the deep tropics (15°N–15°S), UT clouds have a net radiative heating effect on the troposphere from 250 hPa downward by about 0.3 K day$^{-1}$, and this radiative heating enhances the column-integrated latent heating by about 22 ± 3 % (Li et al., 2013; Stubenrauch et al., 2021). Regionally and temporally, this enhancement however varies. It was shown that both ACRE and LP depend on surface temperature (Hartmann and Larson, 2002; Cesana et al., 2019) and column humidity (e.g., Bretherton et al., 2004; Holloway and Neelin, 2009; Masunaga and Bony, 2018; Needham and Randall, 2021; Masunaga and Takahashi, 2024). Therefore, we explore the relationship between latent and radiative heating as a function of these environmental factors (section 5.1) and then, more specifically, for mesoscale convective systems (section 5.2). Since we found the CIRS-ML LP at 1:30 AM/PM more similar to the LP of the diurnally sampled TRMM-SLH over ocean, we consider in the following only precipitating clouds over ocean.

## 5.1   Relationship between LP, ACRE and environment

On average, the radiative enhancement ACRE increases with LP, as seen in Fig. 10a and 10b (black curves) and as already discussed by Stephens et al. (2024). This increase flattens for LP larger than about 250–500 W m$^{-2}$. Figure 10a and 10b also show the relationship between ACRE and LP for different environmental conditions: (1) warm and humid, (2) warm and dry, (3) cool and humid, (4) cool and dry. These environmental conditions are given by SST and by the integrated column water vapour (CWV), obtained from the ERA-Interim meteorological reanalyses.

When stratified by environmental condition, we still observe the increase of ACRE with increasing LP, but for each LP interval the average ACRE is largest for warm and humid situations and smallest for cool and dry situations. Figure 10a (all

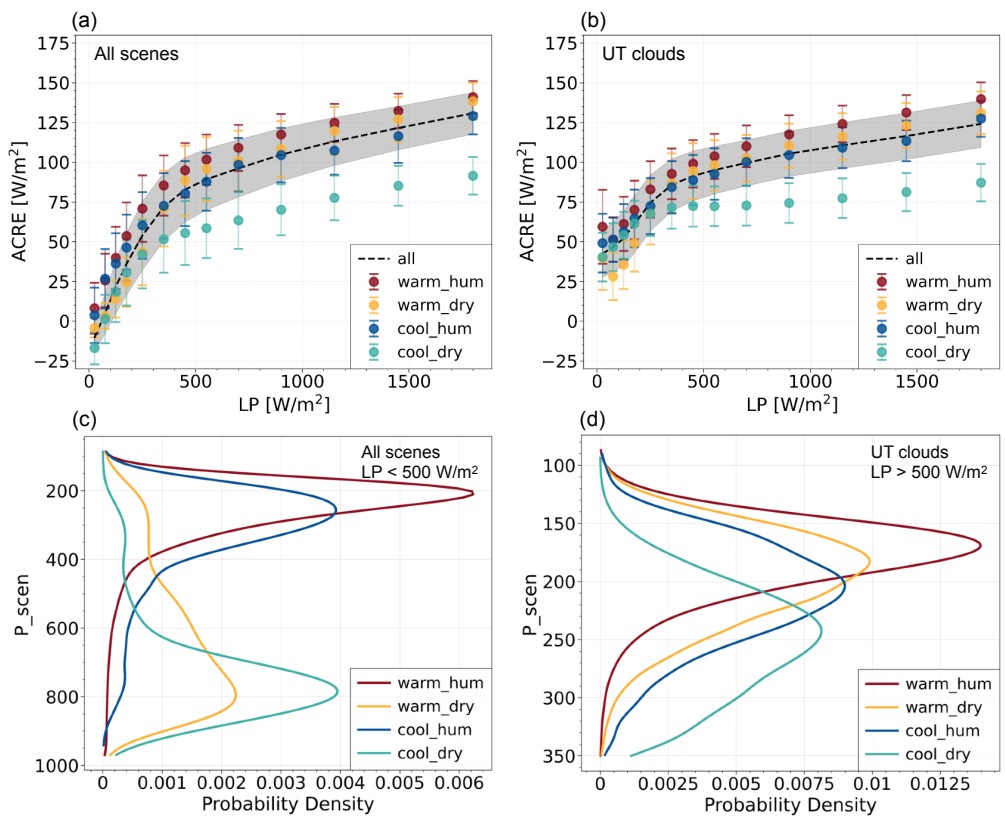

**Figure 10.** ACRE as a function of LP of precipitating cases (a) for all cloud scenes and (b) for UT clouds only for four specific environments (cool-warm, humid-dry). Shaded areas and error bars indicate ±67.5 % of the standard deviation. Probability density of cloud pressure for these different environments (c) for all cloud scenes and (d) for UT clouds, respectively. The environments are defined as: (warm-humid) SST > 302 K and CWV > 60 mm, (warm-dry) SST > 302 K and CWV < 45 mm, (cool-humid) SST < 298 K and CWV > 60 mm, and (cool-dry) SST < 298 K and CWV < 45 mm. All data are from the period 2004–2013, within 30°N–30°S, at a spatial resolution of 2.5° × 2.5°, over ocean.

precipitating clouds) and 10b (only precipitating UT clouds), differ mostly for smaller LP as lower clouds do not produce as much LP. The averages of ACRE in dry situations at these smaller LP show that UT clouds radiatively heat the atmosphere, while low-level clouds cool the atmosphere (aloft). For UT clouds with LP greater than 500 W m$^{-2}$, ACRE differences between cool-dry and warm-humid conditions increase from 30 to 50 W m$^{-2}$ with increasing LP. It is interesting to note that ACRE is similar between warm-dry and cool-humid conditions, and both are quite close to ACRE in warm-humid situations. This means that for UT clouds releasing large latent heat, the SST has a large impact on ACRE in dry than in humid environments.

Now how can UT clouds which release a similar latent heat differ so much in ACRE? The cloud height in these different environments plays a key role, as shown by the cloud top pressure distributions in Fig.10c and 10d, respectively for all precip-

itating clouds with small LP (< 500 W m$^{-2}$) and only precipitating UT clouds with large LP (> 500 W m$^{-2}$). A larger ACRE can be explained by a higher cloud height (lower cloud top pressure): The highest clouds (peak at 170 hPa or about 15 km) are situated in a warm and humid environment and the lowest UT clouds (peak at 240 hPa or about 11 km) are situated in a cool and dry environment, the cloud height distributions in cool-humid and warm-dry environments are situated between those in warm-humid and cool-dry environments, with each being progressively lower than in the warm-humid environment.

For clouds which only release small latent heat, humidity seems to be more important than SST to find slightly larger ACRE (Fig. 10a), as already pointed out by Needham and Randall (2021). This can be explained by the fact that UT clouds are much more frequent in humid than in dry regions, while lower-level clouds exist mostly in dry regions (Fig. 10c).

Why does cloud height differ under varying SST–CWV conditions? Humid environments increase the buoyancy of convective clouds, which allows clouds to reach higher heights (Holloway and Neelin, 2009). In contrast, in dry environments, the lower water vapor content results in smaller plume buoyancies, limiting convection and preventing clouds from reaching the same height as in humid conditions, confirmed by Fig. 10c. The impact of lower tropospheric moisture on buoyancy through entrainment seems to be particularly significant compared to other mechanisms, though additional processes may also contribute (Derbyshire et al., 2004). Under lower humidity conditions, higher surface temperatures lead to higher Convective Available Potential Energy (CAPE, Seeley and Romps, 2015), which provides enough energy to lift air upwards, forming taller clouds. This explains why cloud heights and ACRE in humid and warm-dry conditions are similar, while both cloud heights and ACRE are significantly lower in a cool-dry condition (Fig.10b and 10d). In addition, low-level wind shear can also influence cloud development: moderate low-level wind shear, where cold pool outflow balances environmental shear, can help convective clouds develop to greater heights and persist longer, while too weak or too strong low-level shear tends to suppress deep convection and reduce cloud top height (Rotunno et al., 1988).

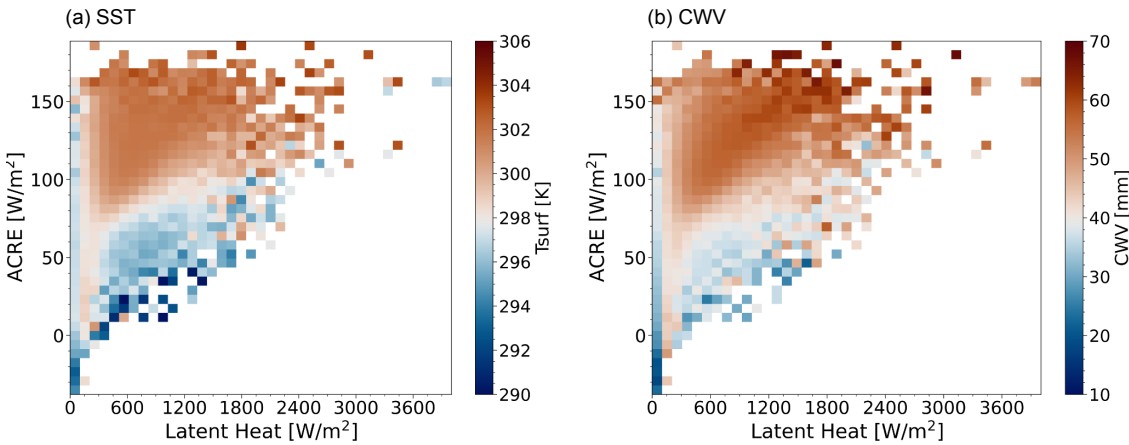

**Figure 11.** Averages of (a) SST and (b) CWV as function of LP and ACRE released by precipitating UT clouds over the ocean, for the period 2004-2013, spanning 30°N to 30°S, at a spatial resolution of 2.5°. LP and ACRE are from CIRS–ML, while SST and CWV are from ERA-Interim at 1:30 AM/PM local time. Each square corresponds to an interval of 100 W m$^{-2}$ in ML LP and 5.75 W m$^{-2}$ in ACRE.

When considering the distribution of UT clouds in the LP–ACRE plane, with SST and CWV averaged over each interval in LP and ACRE, as shown in Fig. 11a and 11b, respectively, we observe a very large spread in ACRE for small LP, which gradually is reduced towards larger LP. As one expects that UT clouds heat the atmosphere, the occurrence of negative ACRE values for LP values less than 500 W m$^{-2}$ should correspond to thin cirrus with lower precipitating clouds underneath. In this case, the CIRS cloud retrieval itself only provides the properties of the uppermost cloud.

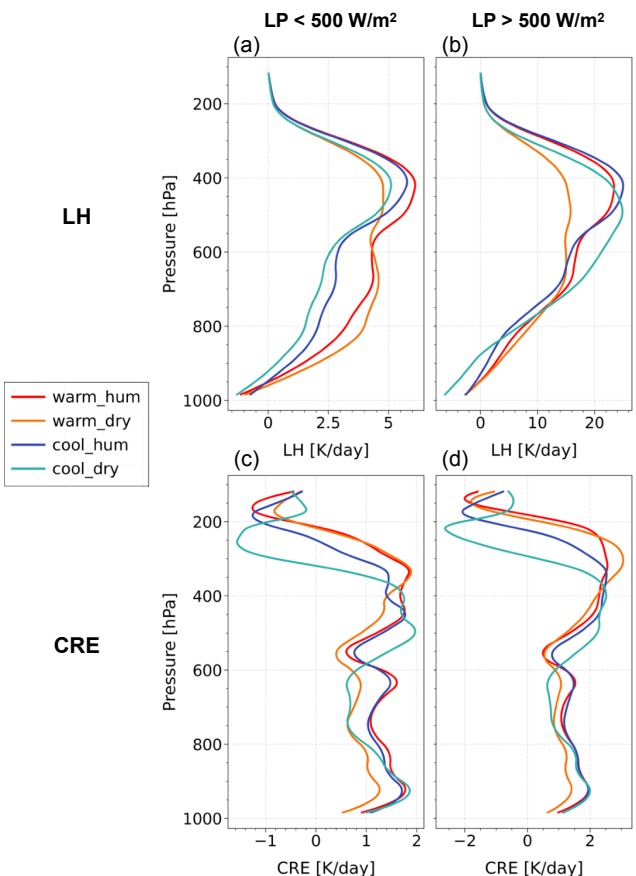

**Figure 12.** LH (top panels) and CRE (bottom panels) profiles of precipitating UT clouds, segmented by an LP threshold of 500 W m$^{-2}$ and averaged across the environments of Fig. 10. All data are from the period 2004–2013, within 30°N–30°S, at a spatial resolution of 2.5° × 2.5°, over ocean.

The occurrences in the LP–ACRE plane and their associated average environments can be compared to the results of Masunaga and Takahashi (2024, Fig. 6): They have characterized three convective regimes (bottom-heavy, mid-heavy, and top-heavy) based on the net moisture and moist static energy (MSE) transport associated with vertical motion (their Fig. 5) and then associated different parts of the ACRE–LP space to these convective regimes, again according to import and export of net moisture and MSE (their Fig. 6). By comparing these convective regimes projected to the LP-ACRE plane with Fig. 11a and 11b, UT clouds in the more or less warm-humid conditions correspond mostly to a mid-heavy convective regime, while the bottom-heavy convection regime, with small LP and a widely spread ACRE, occurs in a cool-dry environment, with UT clouds on top of lower convection. For larger LP, the cases of low ACRE should correspond to the top-heavy convective regime, and these are also associated to a cool-dry environment. The results are robust, when averaged over 5° (Fig.S14).

In order to explain the smaller ACRE in the cooler and drier environment, which corresponds to 'top-heavy' convective regimes, Fig. 12 shows the heating profiles of the UT clouds for the environments defined above, in two LP intervals, with LP > 500 W m$^{-2}$ and LP < 500 W m$^{-2}$. Indeed, for the small LP interval and the cool environments, the LH profiles seem to be dominated by stratiform rain, with a relatively narrow LH peak around 410 hPa, while the LH profiles for the warmer environments also show a heating in the mid- and lower part of the troposphere. For larger LP, the LH profiles are more similar

between these different environments, with an increasing and broadening of the peak towards the mid- and lower troposphere. This indicates either an increase of the latent heating produced by the convective cores (mid-heavy) or additional heating by the bottom-heavy lower convection underneath the anvils (as suggested by Masunaga and Takahashi (2024)). Note that the scales of LH in Fig. 12a (-2 to 6 K day$^{-1}$) and Fig. 12b (-5 to 25 K day$^{-1}$) are different, reflecting the large difference in the LH peak values between the two LP intervals. The relatively small ACRE in the cool-dry environment in Fig. 11 corresponds

to a radiative heating below a height of 450 hPa and a relatively large and broad cooling above this height. The reason why ACRE is smaller under these environmental conditions is that the UT clouds are lower in height (as shown above), and that the cooling above them is more pronounced. The larger cooling can be explained by the fact that lower clouds may be optically denser, in addition to their warmer temperature, thus they emit more LW radiation and the atmosphere above cools more. The larger cooling also leads to a slightly larger vertical gradient in radiative heating. The smaller height may be interpreted as

anvils of convective systems having descended at a later stage of their life cycle (Strandgren, 2018) or as relatively thick clouds with diffusive tops, for which the retrieved (radiative) height may be deeper within the cloud because of very small ice water content in the upper part of the cloud (e.g., Liao et al., 1995; Stubenrauch et al., 2010, 2017).

## 5.2  Diabatic heating of MCS

Deep convection in the tropics leads to a large outflow of anvil clouds. The radiative heating of these UT clouds originating
from convection enhances the latent heating associated with precipitation and thereby strengthens the circulation (Stephens et al., 2024). In the following, we explore the relationship between LH and Q$_{rad}$ within the CSs and MCSs.

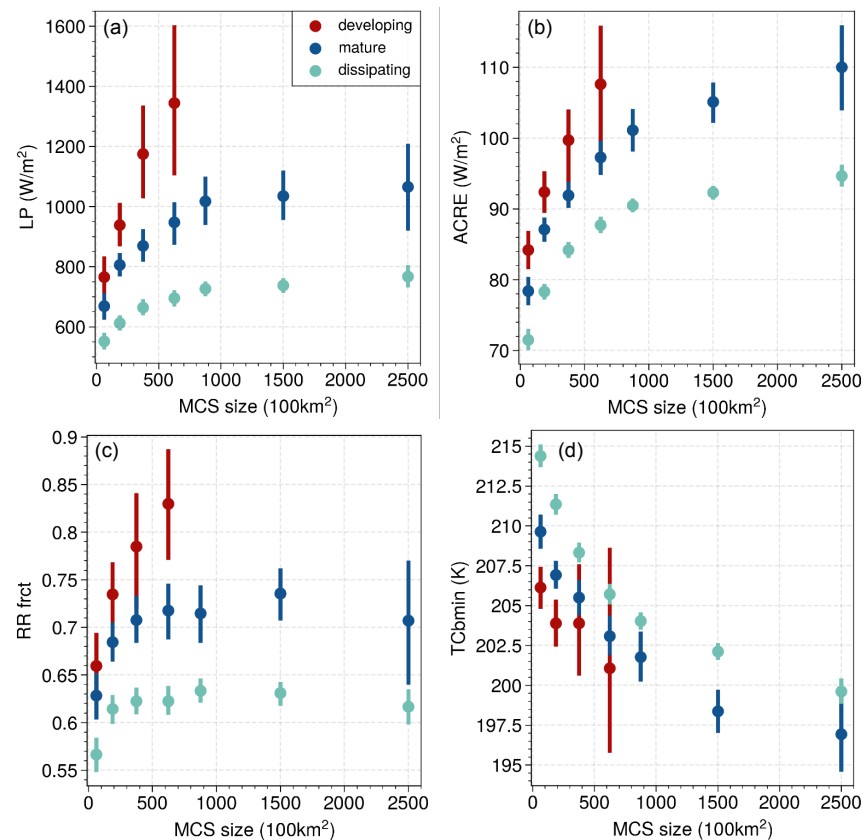

**Figure 13.** LP (a), ACRE (b), rainy area fraction (c) and minimum cloud top temperature within the convective core (d), as a function of the CS size, separately for developing, mature and dissipating CSs (distinguished by convective core fraction: 0.6–0.9, 0.4–0.6 and 0.2–0.4, respectively). All size intervals except the first two include exclusively MCSs. Choosing the coldest part of the core avoids to include parts of the thicker anvil. CSs over ocean, 1:30 AM/PM local time, 2004–2018, 25°N–25°S.

First, we study how LP and ACRE change with the size of the MCS. It has been shown that the horizontal extent of the CS depends on the intensity and organization of convection, but it also changes during its life cycle (e.g., Machado et al., 1998; Takahashi and Luo, 2014; Protopapadaki et al., 2017). Therefore, we analyze the CS / MCS properties separately at different life stages: developing, mature and dissipating, defined by convective core fractions of 0.6–0.9, 0.4–0.6 and 0.2–0.4, respectively. Figure 13 shows that both LP and ACRE increase with MCS size, as expected from Fig. 9. The increase flattens for larger MCS size. Furthermore, for a similar size, LP and ACRE decrease from developing towards dissipating stage, as expected (e.g., Bouniol et al., 2021; Takahashi et al., 2021; Elsaesser et al., 2022). These behaviours are in line with those of the fraction of precipitation area within the MCS and the minimum cloud top temperature within the convective core, respectively: The fraction of precipitating area increases similarly as LP with CS size, and the core top temperature decreases

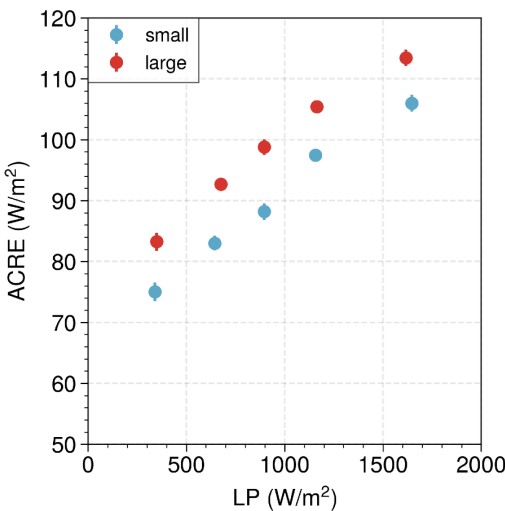

**Figure 14.** Relationship between radiative enhancement ACRE and precipitation intensity LP, averaged per MCS, separately for small MCSs (below the 40th percentile) and large MCSs (above the 60th percentile), considering MCSs larger than 4 grid cells ($1° \times 1°$). Only mature MCSs (convective core fraction 0.4–0.6) are included. MCSs over ocean, 1:30 AM/PM local time, 2004–2018, 25°N–25°S.

with CS size, which explains the increasing ACRE. On the other hand, the slightly decreasing anvil emissivity (not shown) should dampen the ACRE increase.

As deduced from Fig. 6, the reliability of the values of LP average over the CS should increase with increasing CS size. This means that LP should be slightly more underestimated in the first size intervals than in the latter ones. Nevertheless, the results

of Fig. 13 a seem at least qualitatively to agree with the expectations.

In order to isolate the effect of convective organization, we only select mature MCSs and compare in Fig. 14 ACRE between smaller, less organized and larger, more organized MCSs at similar average rain intensity (using vertically integrated latent heating LP as a proxy). On average, the ACRE increases with LP, when both averaged over mature MCS, for both small and large MCSs. Most importantly, we observe for each rain intensity interval, given by LP, that the mean ACRE of larger, more

organized MCSs is by about 10 W m$^{-2}$ larger than the one for smaller MCSs. Considering Fig. 6, we expect LP underestimated by a certain factor, and this factor should be smaller for large MCSs than for small MCSs. Such an underestimation means a stretching of the LP-axis, but this stretching should be slightly larger for the small MCSs than for the large MCSs, while their ACRE would not change. So, indeed, the result in Fig. 14 is robust, and the enhancement effect by convective organization may be even slightly underestimated.

The LP intervals shown in the legend of Fig. 15 correspond to the first, third, and fifth of Fig. 14. As expected (e.g., Houze, 2004), the difference in shape of the LH profiles averaged of the mature MCSs between the larger, more organized MCSs and the smaller MCSs can be explained by a larger contribution of stratiform rain in organized MCSs, with a larger peak in the upper troposphere and a larger vertical gradient down towards the surface from the anvil heating and cooling below, except for the most precipitating ones which show a large heating through the whole atmosphere. It needs more detailed studies at a better

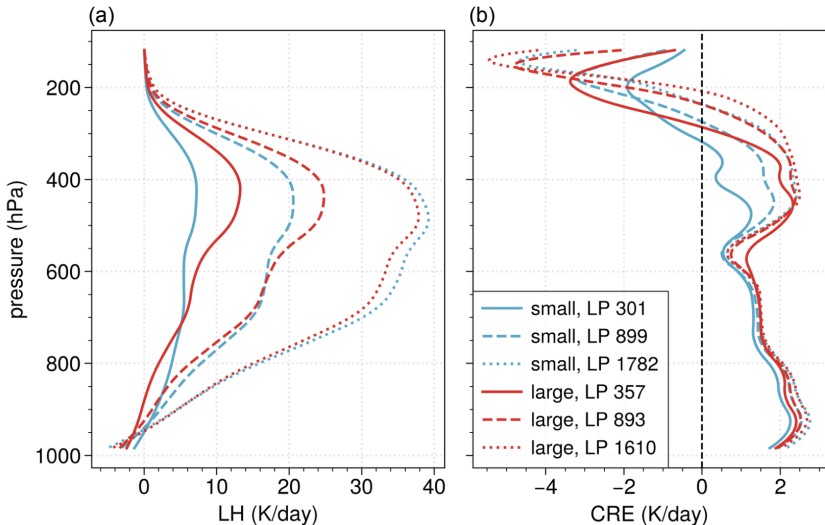

**Figure 15.** Profiles of latent heating (a) and of cloud radiative effect (b) averaged per mature MCS (core fraction 0.4–0.6), shown separately for three LP intervals used in Fig. 14 (lowest, middle, and highest), with mean values of approximately 320, 900 and 1700 W m$^{-2}$ (LP means in the legend in W m$^{-2}$), and also separately for small MCSs (below the 40th percentile) and large MCSs (above the 60th percentile), considering MCSs larger than 4 gris cells (1° × 1°). MCSs over ocean, 1:30 AM/PM local time, 2004–2018, 25°N–25°S.

spatial resolution to understand if the heating in the middle troposphere comes from more productive cores or from anvil with precipitating systems underneath, as already suggested by Masunaga and Takahashi (2024).

In general the core top temperature decreases slightly with increasing LP, slightly more for the larger MCSs (not shown), which is also reflected in the shapes of the radiative heating profiles of these mature MCSs in Fig. 15b. Figure 15b reveals that the vertical gradients increase with increasing LP for both small and large MCSs, with increasing cooling above and increasing heating within the MCSs. Furthermore, in each of the three chosen LP categories, the vertical gradient of the larger, more organized systems is larger compared to the smaller MCSs. This additional ACRE and larger vertical heating gradient then may further support stronger and more sustained convective intensity by enhancing updrafts, maintaining the system, and modifying the larger-scale environment.

## 6  Conclusions and Outlook

In this article we explored the relationship between latent and radiative heating in the tropics. The diabatic heating rate profiles have been obtained from active instruments, which have a sparse sampling. In order to expand these heating rates in space and time, we used techniques based on artificial neural networks (ANN). While the radiative heating and a precipitation classification have already been expanded earlier (CIRS–ML dataset, Stubenrauch et al., 2021, 2023), using ANN networks separately trained for different CIRS cloud scenes with CloudSat-CALIPSO FLXHR data and CloudSat PRECIP-COLUMN,

respectively, we presented here the extension of the latent heating rates. After having assessed the consistency of the extended LH and $Q_{rad}$ fields, we examined the relationship between latent and radiative heating for different atmospheric environments and within MCSs.

For the expansion of the TRMM-SLH latent heating rates (Shige et al., 2009), we used similar ANN regression methods and inputs as for the radiative heating expansion. However, the ANN networks were separately trained for different CIRS-ML
precipitation scenes with gridded data at a spatial resolution of 0.5°. While for the radiative heating rates the predicted averages as well as their variability match well those of the CloudSat–CALIPSO FLXHR target data, the predicted LH profiles agree very well in the means with the TRMM-SLH target data. However, the range of the predicted values is much narrower than the one of the target data, likely due to insufficient input information and its coarse spatial scale. We could further demonstrate that some of the noise in the prediction comes from discrepancies in the rain fraction between TRMM and CIRS-ML (originally
CloudSat), with different instrument sensitivities and spatial sampling. When comparing vertically integrated LH (LP), this noise may lead to an overestimation of LP for small TRMM-SLH values and an underestimation of LP for large TRMM-SLH values.

We reconstructed the 3D latent heating fields at four observation times, at 1:30 AM/PM LT (AIRS) over the period 2004–2018 and at 9:30 AM/PM LT (IASI) over the period 2008–2018.
The zonal averages of LP at 1:30 AM and PM LT align well with those from the full diurnal sampling of TRMM–SLH over ocean. However, over land, the daily mean LP is underestimated because the strong convection in the late afternoon is not captured at these observation times. The observation times of IASI (9:30 AM/PM) underestimate the daily mean LP over both ocean and land. Therefore, we have performed all relationship analyses of the diabatic heating fields using CIRS–ML-AIRS over ocean. The slopes between the monthly averages of TRMM–SLH LP and CIRS–ML LP over ocean increase from 0.54 to
0.82 for scales between 1° and 10°, with slopes of 0.68 and 0.76 for 2.5° and 5°, respectively. Though the complete collocated TRMM-CIRS datasets have less than 5 % of the statistics of the expanded dataset, which may lead to an inhomogeneous sampling, the comparison of the zonal LP averages between TRMM-SLH and CIRS-ML shows indeed a slight underestimation in bands of strong precipitation, like the ITCZ, and a very slight overestimation in bands including deserts.

Geographical maps show a close association between LP and the vertically integrated, atmospheric cloud radiative effect
(ACRE) and reflect well the characteristics of ENSO events. Furthermore, the horizontal structure of LP closely matches the one from precipitation obtained from TRMM and GPCP, though the range of the CIRS-ML LP values is underestimated. Nevertheless, when keeping these systematic biases into account, this dataset can be used to study the horizontal structure of LP at four specific observation times. This is not possible with TRMM monthly averages at a specific time because they do not fill at all the tropical band.
The main purpose of this article was to study the relationship between latent and radiative heating for UT clouds, first under different atmospheric environments characterized by sea surface temperature (SST) and column humidity (CWV), and then within MCSs. For a similar rain intensity, given by LP, ACRE is generally largest in warm-humid conditions and smallest in cool-dry conditions, which is essentially linked to higher and lower cloud height, respectively. For UT clouds releasing large

latent heat, SST has a larger impact on ACRE in dry than in humid environments. On the other hand, humidity plays a larger role in cool environments.

The distribution of UT clouds in the LP–ACRE plane shows a large spread in ACRE for small LP, which is gradually reduced towards larger LP. Compared to the association of convective regimes to the LP-ACRE plane by Masunaga and Takahashi (2024), the UT clouds in the more or less warm-humid conditions correspond mostly to mid-heavy convective regimes. The cool-dry environments are linked on one hand to the bottom-heavy convection regimes with small LP and a widely spread ACRE, and on the other hand, for larger LP and smaller ACRE, to the top-heavy convective regimes. The smaller ACRE can be explained by a slightly larger cooling above the clouds and a smaller (radiative) height of these clouds.

Comparing MCSs of similar size, both mean LP and ACRE decrease from the developing towards the dissipating stage of the MCSs. Furthermore, both mean LP and ACRE slightly increase with MCS size. In order to study the effect of convective organization, we selected MCSs in the mature state and compared the relationship between ACRE and LP separately for small and large MCSs, and we found an ACRE enhanced by about 10 W m$^{-2}$ for larger, more organized MCSs than for smaller, less organized MCSs at similar average rain intensity. Convective organization also increases the vertical gradient of the mean radiative heating of these systems at similar rain intensity (LP). This enhanced ACRE and larger vertical heating gradient then may further support stronger and more sustained convective intensity by enhancing updrafts, maintaining the MCS, and modifying the larger-scale environment. As expected, the shapes of the LH profiles of mature MCSs show that larger, more organized MCSs have a larger contribution of stratiform rain than the smaller MCSs.

Future studies should consider also the environment around the MCSs and in particular the time dimension. The latter can be achieved by combining the CIRS–ML heating rates with deep convective cloud systems, using a better spatial and temporal resolution (Fiolleau and Roca, 2013; Takahashi et al., 2021), providing additional parameters such as their life stage, lifetime, and maximum size during their lifetime. The distribution of the UT clouds and their associated environment in the LP–ACRE plane can also be used to evaluate climate simulations, at least qualitatively.

The ML method for the expansion of TRMM LH rates may be further improved undertaking an ANN training using collocations at the scale of the AIRS / IASI footprint and integrating a substructure of measurements at an even smaller scale like from the Moderate Resolution Imaging Spectroradiometer (MODIS) together with an auto encoder method as in Shamekh et al. (2023).

*Data availability.* The TRMM latent heating rates used in this study are from the Tropical Rainfall Measuring Mission (TRMM) dataset: GPM PR on TRMM Spectral Latent Heating Profiles L3 1 Day $0.5° \times 0.5°$ V06, provided by the Goddard Earth Sciences Data and Information Services Center (GES DISC), available at https://doi.org/10.5067/GPM/PR/TRMM/SLH/3A-DAY/06 (Shige et al., 2009). The CIRS-ML radiative heating rates, vertical cloud structure, and rain rate classification data are from the Global Energy and Exchanges Process Evaluation Studies (GEWEX PROES), accessible at https://gewex-utcc-proes.aeris-data.fr/. The TRMM (TMPA) Precipitation L3 1 Day $0.25° \times 0.25°$ V7 dataset, used for the comparison of LH horizontal structure with ML-expanded LH, is also available from GES DISC (https://disc.sfc.nasa.gov/datacollection/TRMM_3B42_Daily_7.html). The Global Precipitation Climatology Project (GPCP) Monthly Precipitation Climate Data Record (CDR) is provided by NOAA's National Centers for Environmental Information (doi:10.7289/V56971M6).

*Author contributions.* XC developed and evaluated the ANN models for the expansion of LH, performed the analysis of the expanded LH and $Q_{rad}$ data, wrote the main sections of the manuscript, and produced the figures. CJS conducted the cloud-system-related analysis, wrote Section 5.2, generated the corresponding figures, and made substantial contributions to optimizing and revising the manuscript. GM analyzed the zonal averages of vertically integrated diabatic heating, produced Fig. 5 in Section 3, and contributed to the overall improvements of the manuscript.

*Competing interests.* The authors declare that they have no conflict of interest.

*Acknowledgements.* We gratefully acknowledge the efforts and collaboration of the AIRS, CALIPSO, CloudSat, IASI, and TRMM science teams in providing the datasets used in this study. We also extend our thanks to the engineers and space agencies whose work ensures the consistent quality of these data. The computational resources provided by IPSL ESPRI Mesocenter were instrumental to the completion of this work. We also appreciate the technical support provided by Jeremie Trules during the transition to a new remote server. Special thanks go to Camille Risi, Hirohiko Masunaga, and Laurent ZX Li for their insightful discussions, and to three anonymous reviewers for their valuable feedback, which greatly improved the clarity and precision of this manuscript. Finally, we would like to acknowledge the use of Fabio Crameri's "Scientific Colour Maps" (Crameri et al., 2020) in the preparation of some of our figures, which adhere to principles of readability and accessibility in scientific visualization. This work has been supported by the China Scholarship Council (CSC) program (Grant No. 202208070008), the French National Centre for Scientific Research (CNRS) and the Centre National d'Études Spatiales (CNES).

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
