# Peer review of "Relationship between latent and radiative heating fields of Tropical cloud systems using synergistic satellite observations"

_EGUsphere, 2024_

## Editor Comment (EC1)

Relationship between latent and radiative heating fields of Tropical cloud systems using synergistic satellite observations

Xiaoting Chen, Claudia J. Stubenrauch, and Giulio Mandorli

This paper is to use synergistic satellite data from active instruments; it also applies artificial neural network regressions on InfraRed Sounder data, and meteorological reanalyses to investigate the relationship between **latent heating (LH)** and **radiative heating (RH)** for **mesoscale convective systems (MCS).**

The main results show (1) the zonal averages of vertically integrated LH (LP) at 1:30AM and PM LT align well with those from the diurnal sampling of TRMM–SLH LH over ocean; (2) the surface temperature has a larger impact on the *atmospheric cloud radiative effect* **(ACRE)** in dry than in humid environments for **Upper Tropospheric** (UT) clouds; (3) humidity plays a large role in enhanced ACRE for lower clouds, producing relatively small latent heat; (4) the mean ACRE per MCS increases with LP; and (5) LH profiles of mature MCSs *have a larger contribution of stratiform rain* than the smaller MCSs,

This paper is interesting because it applies artificial neural network to investigate the relationship between LH and RH.

I have comments that the authors need to address to have paper to be published.

**Comments**
Line 17: Suggest deleting "the precipitating parts of".

Line 25: Suggest deleting "closer".

Line 32, tropical: May identify the area of "tropical: (i.e., 30º or 20º south to north).

Line 83; Hagos *et al*.: Not sure if this reference needed.

Line 85, continuous: Not sure what it means.

Line 87: Please check the year of Shige et al. (2004 or 2003).

Line 94-95: Tao *et al.* (2022) did not state (or show) the comparison between SLH derived heating and re-analyzed heating profile. The CSH and SLH derived LH shown in Hoges et al. (2010) are from old version of CSH and SLH retrieved LH. The new version for both SLH and CSH is V6 (shown in Tao *et al.* 2022).
In addition, the reanalyzed is a combination of model and observation. That is why different re-analyzed LH (shown in Hoges *el.* 2010) are different.

Line 95: It is Shige *et al.* (2007) that paper described the SLH algorithm.

Line 99-128 (section 2.2): What is the horizontal and vertical resolution (0.5 x 0.5 degree and 50 layers) of these three-satellite derived cloud information (AIRS, IASI and CIRS)?

ERA interim's horizontal resolution is 0.75 x 0.75 degree. Could you please describe what are spatial resolution of these three-satellite derived cloud information? Does *cubic spline function* also apply to the satellite cloud information (0.5 x 0.5 or 075 x 0.75degree)?

Line 135: What is sub-grid structure? Is it for making all data to 0.5 x 0.5 degree and 10 vatical layers?

Line 144-145: Would it be nice to also use TRMM/GPM derived rainfall intensity.

Fig. 1: What do dark, and light blue color represent?

Line 160: What do you mean the "maximum" 0.0 mm h$^{-1}$?

Line 162-166: The "%" is for frequency (or area coverage). The non-precipitation means no surface rain. Correct?

Fig. 2: The LH profiles are from SLH algorithm. Correct? In addition, please just refer one specific SLH paper that describes the algorithm design. The figure uses two different scale for LH (from -10 to 40) and RH (-10 to 30). Suggest using the same scale for both LH and RH.

Line 160-211; section 2.5.1: It may be a good idea to have a schematic diagram that shows the design of ANN for predicting LH/RH. Maybe a good idea to **show the key parameters for predicting LH** in diagram.

Would ANN-LH predict certainty or uncertainty?

Line 185: What is impact on "randomly divided" on the retrieved product?

Line 186: What is the validation data? (Is it SLH derived LH?)

Line 215: How are the convective and stratiform rain classified (from TRMM or else)?

Line 223: What is the "true data"?

Fig. 3: RH is for longwave (not total radiative heating/cooling). Why is only LW shown?

Line 230: How do the meting processes affect longwave cooling?

Figs. 2, 3 and 4: Please also plot/show total and UT LH in Fig. 3 (for comparison with LH showing in Fig. 4). In addition, please plot/show the LH over ocean vs land as those shown in Fig. 4.

Line 267: Change minor to small (also in other places).

Fig. 5: What is LP from TRMM? Is it SLH derived LH? Is GPM data used?
Please use the term TRMM-LSH (not TRMM). Please also change TRMM to TRMM-LSH in the text and figure caption.

Fig.6: Please elaborate why LH/RH is only from 10 S to 10 N). Why does not show, 30S – 30, as shown in other figures?

Line 292-293: Please elaborate more details (why ANN does not capture extreme event; also, what is "well" extreme events). Also need to use TRMM-SLH (not TRMM) in the statement

Line 298: Change Figure 7a-d to Figure**s** 7a-d (or change show to shows).

Line 312-313: Why is there "less convective activity, there is more low-level cloud formation"?

Fig. 7: What is 1h 30 AM/PM? (at the end of caption).

Figs. 7 and 8: Suggest quantifying the similarity and difference with statistical analyses.

Line 343: What is "less" 26,000 MCSs? Please also show the global – geographic distribution of these MCSs. Are only convective cores in these MCSs considered? Usually, there are stratiform cloud (generally with large area coverage than that of convective core). Is it possible to estimate stratiform % in these MCSs??

Fig. 9: Why is there large/small cooling within the Fig. 9a?

Line 352, -7.5 K Day$^{-1}$; Line 356, 40 K Day$^{-1}$: It is supposed to heating (release heating from condensation/deposition) in the convective core.

Line 357: Change "produce" to "show".

Line 363-364: It is not clear, what is *intensity* is directly related to heavy precipitation?
What is heavy precipitation (do you mean precipitation event)?

Line 398: Change "produce" to "release".

Line 405: Change "greater" to higher.

Line 404-409: There are other dynamic factors "i.e., low-level wind share, CAPE" that can play important role on cloud development.

Line 412: Does humidity have impact on atmospheric radiative effect? (Heat atmosphere could reduce relative humidity).

Line 418, 420: What is mid-heavy, bottom-heavy, and top-heavy convective regime?

Line 419: Where is "on top of lower convection"?

Line 423, two LP intervals:  Need to mention that " the two regimes are with LP > and < 500 W m$^2$.]".

Line 424-425: Please elaborate in detail on the statement: the LH profiles seem to be dominated by stratiform rain, with a relatively narrow LH peak around 410 hPa.
This maybe the 1$^{st}$ time that the authors mention the stratiform rain.

Fig. 12: The scales used in Fig. 12a (-2 to 6) and Fig.12b (-5 to 25) are quite different.  Please mention this in the text.

Line 436-438, life cycle:  Can you justify the discussions on "can ANN produce the life cycle information"?

Figs. 13 and 14 caption: Is 1:30 AM or PM local time?  Also, why not comsider30 S to 30 N (as some of other figures) in the discussion?

The definition of developing, mature and dissipating stage of MCS need to be elaborated in detail (or refer to observation or show the structure – both vertical and horizontal for these)

Line 449: What is the "minimum" temperature within the convective core?

Line 456, 0.35 compared to 0.60; Both are classified as mature stage. Why do you need to compare these two mature stages?

Line 461, 20 km$^2$: I thought 0.5 degree is horizontal resolution.  Where is 20 km$^2$ for small rainfall intensity from?

Fig. 15: What is the LP 212, LP 648, LP 1513, LP 255, LP 682 and LP 1357 (within the figure)?
No discussion

Line 471-473 and Fig. 15(a): Please elaborate the followings
"Larger, more organized MCSs have a larger contribution of stratiform rain than the smaller MCSs, except for the most precipitating ones which show a large heating through the whole atmosphere".  It is not clear where this information from Fig. 15(a)

---

## Author Comment (AC1)

**Author response to Referee #1:**

Dear Referee,

We sincerely thank you for your thoughtful and constructive comments, which have greatly helped us improve the manuscript. Below, we provide our point-by-point responses to each comment. All changes made to the manuscript according to the comments of the three reviewers have been carefully highlighted in the revised version.

**General comments:**

Latent heating and radiative heating are two major diabatic heating sources of the atmosphere. However, the observations of global latent and radiative heating profiles are difficult to obtain. Active sensors onboard satellites provide a way to measure these profiles, such as the products from TRMM and GPM. However, TRMM and GPM satellites are low-orbit satellites thus the spatiotemporal coverage of the heating profile data are limited. The previous work from the author(s) have applied a ML technique to extend the radiative heating profile to greater coverage. This paper does similar thing but for latent heating (LH), with further analysis of LH profiles for different surface conditions, different environmental conditions and MCS characteristics. One of the main conclusions is that although the mean values show good agreement, the ML-expanded latent heating profiles show much smaller variability than the target data. This is different from the ML-expanded radiative heating profiles. The authors made a further analysis and suggested that the data are only robust for coarse-resolution larger than 2.5deg.

Although some results of this paper are interesting and this expanded dataset would be useful for scientific community, I feel that the values and limitations need to be carefully elaborated and better demonstrated. One major concern I have is that since the authors explicitly suggest that "our ML-expanded LH dataset is suitable at scales larger than about 2.5∘", how robust can we trust the results for MCSs in the size of around 1x1deg, or 100kmx100km, as the authors show in Fig. 13 to 15?

*We thank the reviewer for his very constructive and thoughtful comments.*

*Indeed, it is very difficult to evaluate very precisely this new dataset, since we had only the collocated data for a direct evaluation, with a sparse sampling (which was the reason why we used ANN to expand the data).*

*Most ML datasets are only evaluated using a part of the collocated data (as in our section 2.5.2). The results in Table S1 show relatively small mean absolute errors, of about 0.5 K/day in the case of heavy rain. However, instantaneous predictions do have indeed much larger uncertainties as we have shown that the variability of the predicted values are smaller than the original ones.*

*Therefore, we used the complete TRMM data for a further comparison (section 3.2), but the TRMM data themselves have a very difficult diurnal sampling. We partly eliminated the uneven time sampling problem of TRMM by concentrating on the latitudinal band near the equator. Still, we had the problem to compare specific times with the complete sampling, but since we could show that on average the AIRS observations at 1:30 AM/PM agree well on average with those of the complete TRMM data over ocean, we could build scatter plots of monthly mean values, but still these are monthly means.*

*Further inspired by the reviewer's comment, we have re-collocated the two datasets again, by including variables which were not used for the training (like latitude, longitude, date, fraction of certain rain) as well as the produced CIRS-ML LH, and we used this complete dataset for further investigations.*

*We analyzed the collocated data further, though these consist of these relatively narrow orbit pieces (shown in red in Fig. 1), so that averaging instantaneous values over larger regions do not fill out these*

*regions for daily statistics. By considering the latitudinal means of the collocated TRMM data and the collocated ML data (the data displayed in Fig. 3, but this time for the whole statistics of the collocated data and not only the 20% test data), we show that in regions of large rainfall the ML data underestimate LP and in regions of low rainfall the ML data very slightly overestimate LP. This could already be indirectly derived from the scatter plots in Fig. 6, but this time we compare data taken at the same local time.*

*In order to further understand the noise and biases, we tried to link discrepancies in LP with those in rain fraction over the 0.5° grid cells. Therefore, we compared CIRS-ML certain rain fraction as function of TRMM rain fraction and TRMM-SLH LP (new Fig. S10), and they look very coherent, with an average increase of the certain rain fraction with both, as one would expect. When separating cases with small and large fraction of certain rain within the grid cells, the CIRS-ML LP is much larger for a large fraction, as one would again expect. However, one would not expect cases with a large certain rain fraction for very small TRMM-SLH LP. Indeed, this is true as the average behavior shows a mean close to 0.*

[Figure]

**Figure S10: Average relationships for high-level clouds between (a) CIRS-ML certain rain fraction and TRMM rain fraction, (b) CIRS-ML certain rain fraction and TRMM-SLH LP and (c) CIRS-ML LP and TRMM-SLH LP for all and for cases with CIRS-ML certain rain fraction < 0.5 and > 0.5. Statistics of collocated data over ocean 30°N–30°S, over the time period from 2004–2013.**

*The distributions of CIRS-ML certain rain fraction in the CIRS-ML LP – TRMM-SLH LP space, shown in the new Fig. S11, explain then the noise, in particular for small TRMM LP. So, this noise can be mostly explained by a few individual cases which show a mismatch between the certain rain coverage obtained from ANN classification of certain rain identified over CloudSat samples (1.25 km × 2.5 km), and the TRMM radar samples (5 km × 5 km). Nevertheless, what is important to note is that the CIRS-ML LP seems to be coherent with the certain rain coverage, even though this variable was not used in the training! Considering the relationship of the averages, this leads to a slightly larger CIRS-ML LP compared to TRMM LP for small TRMM LP and a slightly smaller CIRS-ML for large TRMM LP, in addition to a large underestimation of extremely larger values, as expected from the ML technique. We needed to use the rain information from CloudSat, which itself was already expanded by ML, because this is available in the CIRS production.*

*We have improved the text in section 2.3 and added these findings in section 3.2.*

[Figure]

**Figure S11: Averages of CIRS-ML certain rain fraction as function of TRMM-SLH LP and CIRS-ML LP of high-level clouds over the ocean, from collocated data for the period 2004-2013, spanning 30°N to 30°S, at a spatial resolution of 0.5°. Each square corresponds to an interval of 75 W/m² in ML LP and 150 W/m² in TRMM-SLH LP.**

*We understand fully the concern of the reviewer that the MCS results around small MCSs may be biased. Therefore, we have excluded MCSs with a size less than 1°, and instead of a fixed size threshold we use the 40% largest and smallest MCSs. The results become even clearer (new Fig. 14). This shows that the results are robust. In addition, we know that when averaging over larger areas, the LP is less biased and that for small LP there is a slight overestimation and for very large LP there is a large underestimation. This means that the points in Fig. 14 move very slightly to the left for small LP and to the right for large LP, and this more for the small MCSs than for the large MCSs, which means that the effect we show here is even slightly underestimated. So, the conclusion that the ACRE is slightly larger for the same LP (or rain intensity) in the case of larger MCSs is certainly robust!*

*However, we took out the old Fig. 14a which presented a very similar relationship between ACRE and LP for developing, mature and dissipating stages, because since in general the size of the MCSs increases from developing towards dissipating (which means dissipating convection but not dissipating cloud shield) as we know from Fig.12 of (Stubenrauch et al. 2019) or Fig. 9 from (Protopapadaki et al. 2017), and therefore the LP averages over developing MCSs may be stronger negatively biased than the dissipating ones. This would lead then to the conclusion that ACRE for MCSs with similar LP may be slightly larger for dissipating MCSs than for developing ones, but we can't give any quantitative estimate.*

*All in all, we can give qualitative statements about relationships, but it is difficult to quantify the results. Nevertheless, since we have an idea about the direction of the biases, it helps in the interpretation. We included this kind of interpretation into the manuscript.*

**Specific comments:**

- Line 24: as RH is usually referred as relative humidity, maybe change the abbreviation (e.g. Qrad)

  *Thank you for this valuable suggestion. Indeed, this will help clarity. We have incorporated this change and replaced "RH" with "Qrad" throughout the revised manuscript.*

- Line 61: maybe need a little bit more introduction on CRE and ACRE, for example, from which data source did you obtain the CRE/ACRE data (clear-sky and full-sky radiation)? Sometimes the authors state CRE and radiative heating profiles together (even the units in Fig.5 have K/day and W/m2 on the same panel, this confuses me a bit), do they represent similar effect?

*The cloud radiative effect (CRE) is calculated as the difference between all-sky radiative heating rates and clear-sky radiative heating rates, with the unit K/day. The atmospheric cloud radiative effect (ACRE), already defined for example by Li et al. (2015) and Harrop and Hartmann (2016) as the difference in cloud radiative effects between the TOA and the surface, and corresponds to the vertically integrated CRE, with unit W/m².*

*To clarify this point, we have revised the sentence to:*

*"In our analyses, we use the following definitions: LH refers to the latent heating profile; LP denotes the vertically integrated latent heating; Qrad represents the radiative heating profile; CRE (cloud radiative effect) refers to the difference between all-sky and clear-sky radiative heating rates, expressed in units of K/day; and ACRE (atmospheric cloud radiative effect) represents the vertically integrated CRE, with units of W/m²."*

- Fig.1: more details of the figure are needed: what do the different blue colors mean? what are the thin and thick swaths?

  *Taken into account. We have added one sentence in caption of Fig. 1 to:*

  *"The narrow swaths represent TRMM–PR orbits, while the broader swaths represent Aqua–AIRS orbits. Shades of blue indicate variations in sampling time difference."*

- Line 160: maximum should be minimum

  *We changed this to 'with a large peak at…' in the text.*

- Line 162: what are the thresholds in your rain intensity categorization?

  *The thresholds for rain intensity categorization were mentioned in section 2.3, but indeed, only the thresholds used in the rain intensity classification per AIRS footprint were mentioned. We have now completed the description including the propagation to the 0.5° grid cells.*

  *"A rain intensity classification (0 for no rain, 1 for light rain, 2 for heavy rain) was also obtained by ANN classification per footprint, but trained with precipitation rate data from CloudSat (2C-PRECIP-COLUMN, Haynes et al., 2009). The rain intensity classification considers light rain to be =< 5 mm h$^{-1}$ and heavy rain > 5 mm h$^{-1}$. In combination with a CloudSat 2C-PRECIP-COLUMN quality flag, which indicates certain rain, also expanded via a binary ANN classification, a 'rain rate indicator' was constructed and then averaged per 0.5° grid cell (Stubenrauch et al., 2023). The rain intensity classification used for the scene identification for the ANN training and production is based on this averaged rain indicator, with heavy rain starting probably already around 2.5 mm/h. We could show that this category corresponds to an average certain rain fraction of at least 0.8 (not shown), while the light rain category corresponds only to 0.3 over ocean (0.5 over land). The advantage of using the CIRS-ML rain intensity classification, is that the CIRS-ML data are available together with the whole CIRS-AIRS and CIRS-IASI data records, so we can use them for a scene identification for the training as well as for the application of the ANN models developed for these different scenes discussed in section 2.5.1."*

- Line 164: How do you define UT clouds in this paper?

  *High-level clouds are defined in line 118 (previous version):*

  *"CIRS cloud types are defined according to p$_{cld}$ and ε$_{cld}$ as high clouds (P$_{cld}$ < 440 hPa)…"*

*UT clouds are defined in line 331 (previous version):*

*"We consider UT clouds with $P_{cld} < 350$ hPa..."*

*Since the definition of UT clouds is introduced much later, it may cause confusion with high-level clouds. To address this, we added the sentence "UT clouds with $P_{cld} < 350$ hPa are part of the high cloud category." in revised manuscript (line 149).*

*Additionally, there was a confusion in the manuscript. The training of the ANNs was separately done for high-level clouds and mid- and low-level clouds. We carefully reviewed the use of "UT clouds" and "high-level clouds" in both the figures and the text, and made necessary revisions, mostly in section 2.*

- Line 192: I am not an expert of machine learning, but 20 iterations look a lot to me if you don't see any improvement in the loss function. Is it true that it is still not overfitting after 20 iterations without improvement?

  *The appropriate early stopping iteration count depends on the model complexity, dataset size, and training configuration. Since our model is relatively simple, the dataset is large, and the learning rate is small (1e-4), it may take more epochs to see improvement in the validation loss, as for heavy rain shown in Fig. S4. However, also for the scenes for which after 10 iterations no further improvement was seen the validation loss and training loss are similar, indicating no overfitting. So the 20 iterations used here seem to be appropriate.*

- Line 247: Is there any explanation of the two peaks of LH profiles over ocean?

  *The two peaks in the LH profiles over ocean (Fig. 4) at 450 hPa and 850 hPa correspond to the contributions from high-level clouds and low-level clouds, respectively. We have updated line 247 to:*

  *"Over ocean, the mean LH profiles have two peaks, at 450 hPa and at 850 hPa, corresponding to the contributions from high-level clouds and low-level clouds (shige et al., 2004), while over land the LH is mostly produced by high-level clouds."*

- Fig. 5: which variables are the left and right y axes corresponding to?

  *Thank you for pointing this out. The dual y-axes in Fig. 5 both correspond to integrated latent/radiative heating. The values are originally in W/m², which is the correct unit for the integrated quantities. The K/day axis represents the converted heating rate. To avoid confusion, we have removed the K/day axis and kept only the W/m² axis.*

- Fig. 5: why is the SW radiative effect is almost zero? My direct intuition is that when cloud exists, shortwave radiative effect should be pretty negative.

  *In Fig. 5, the SW radiative effect is not the TOA effect but the effect within the atmospheric column, so it should be slightly positive, as the SW heats within the clouds, while indeed the heating is less underneath an opaque cloud than for clear sky (see for example Fig. 3 of Stubenrauch et al. 2021). It is also very small, because the SW ACRE shown is the 24h mean (SW ACRE at 1:30PM weighted by $1/(\pi \times \cos \theta)$).*

  *However, since our main focus is the coherence between TRMM–SLH LP and ML-predicted LP, and not the radiative effect, we have removed the parts on SW and LW radiative effects and the corresponding text to avoid unnecessary complexity.*

- Fig. 5: The 'pink solid line' mentioned in the caption is not shown in the figure.

  *We have revised Fig. 5, keeping the separation ocean – land, and one part went into the supplement to keep the flow of the paper. So the new Fig. 5 shows original TRMM with diurnal sampling and the CIRS-ML production separately for the 4 observation times, and the supplemental figure (Fig. S9) shows the comparison between the CIRS-ML data and the collocated results (for this we needed to rerun the collocation in order to add latitude and longitude to the dataset; this took a little effort, because the machine on which the collocation was run changed in the meantime), separately for all 4 observation times*

  *So, Fig. 5 shows the effect of diurnal sampling, and the Fig.S9 the biases in regions with strong rain (tropical peak region) and in regions with not much rain (subtropics), with a slight underestimation in the first and a very slight overestimation in the latter. The underestimation is largest when the rain is the heaviest, as seen in Fig. 6.*

[Figure]

[Figure]

**Figure 5: Zonal averages of vertically integrated LH (LP) at 4 specific observation times (1h30, 9h30 AM/PM) from the CIRS-ML production and for the original TRMM, including diurnal sampling, (a) over ocean and (b) over land. Latitude intervals are 1°. LP from TRMM data for the period 2008–2013 is represented by a solid black line. The green dashed line and bright green dash-dotted line represent LP from ML regression using TRMM–AIRS (2008–2013) as inputs, at 1:30 AM and 1:30 PM, respectively. The blue dashed line and light blue dash-dotted line represent LP from ML regression using TRMM–IASI (2008–2013) as inputs, at 9:30 AM and 9:30 PM, respectively. Shaded areas correspond indicate inter-annual variabilities.**

[Figure]

**Figure S9: Zonal averages of vertically integrated LH (LP) of collocated data (a) at 1:30 AM and PM over ocean, (b) at 1:30 AM and PM over land, (c) at 9:30 AM and PM over ocean, (b) at 9:30 AM and PM over land. Black solid lines: TRMM LP for AM observations. Gray dashed line: TRMM LP for PM observations. Dark green solid line: LP from ML regression using AIRS or IASI as inputs for AM observations. Light green dashed line: LP from ML regression using AIRS or IASI as inputs for PM observations. Shaded areas correspond to inter-annual variabilities. The latitude intervals are 2°, and the time period is 2008-2013.**

- Line 334: do they need to be neighboring grid cells or only by cloud height to be merged into the same system?

  *As the cloud pressure of UT clouds retrieved by CIRS has an average uncertainty of about 25 hPa, we only merged adjacent grid cells with similar cloud pressure (slightly above the mean error). The UT clouds systems are constructed from neighboring grid cells, containing at least 90% UT clouds ($P_{cld} < 350\ hPa$) and having similar cloud top heights, within a range of 8 hPa $\times In(P_{cld}\ [hPa])$.*

  *We have added one word to the sentence of line 420:*

  *"First, adjacent grid cells with UT clouds of similar height…"*

- Line 335: I don't quite understand this sentence "the size ... is computed... by the number...". Is there anything missing?

  *To clarify this point, we have revised the sentence on line 421 to:*

  *"Then the size of convective cores is determined first by counting the number of grid cells with cloud emissivity $> 0.98$ within regions where the cloud emissivity exceeds 0.93 and then by multiplying this number by the grid cell size of 0.5° (approximately 3000 km$^2$).".*

*We needed to group first regions with slightly smaller emissivity in order to reduce the noise in the determination of the number of cores. If one groups only grid cells with emissivity > 0.98, then one obtains many more multi-cores (see Protopapdaki et al. 2017).*

- Fig. 9: this is another place that I don't quite understand what this paper use, is it CRE or radiative heating?

  *We apologize for the confusion caused by the unclear definitions of CRE and radiative heating earlier. In Fig. 9, it is the CRE being shown, as it represents the net effect of clouds after subtracting the clear-sky radiative heating.*

  *We have already revised the corresponding text in the manuscript to clarify this point.*

- Line 364: not sure I understand this statement. Are you comparing MCS intensity relations with MCS size and opacity? or comparing radiative heating profile with LH profiles?

  *Section 4 was brought in to describe the construction of the mesoscale convective systems. We then thought to illustrate the effect of different proxies for precipitation intensity. Some of these proxies are very indirect, like the size of the systems or the minimum cooling above the precipitating part of the convective system (the LW cooling above the cloud stands for the opacity). For the latter, we select the grid cell within the precipitating part of the convective system with the minimum value of the cooling above the cloud. We compare the latent heating rate profiles with the one for heavy rain MCSs, which is the most direct proxy. Figure 9 was meant to test the coherence of the data. The LW cooling above the core comes from the CIRS-ML data and the core identification from CIRS and using CIRS-ML rain classification. And we can show that the CIRS-ML latent heating profiles change in coherence with the different proxies.*

  *Furthermore, we show that on average the minimum LW cooling above the heavily raining MCS's is the largest. This shows again that the minimum cooling above the participating part of the convective system seems to be an interesting proxy for rain intensity.*

  *We have rewritten the paragraph on the description of these results, and hope that it is clearer now.*

- Line 372-373: from where did you obtain this conclusion that LP is more reliable over ocean?

  *Indeed, the wording was not well chosen. We have rewritten this phrase as:*

  *"Since we found the CIRS-ML LP at 1:30 AM/PM more similar to the LP of the diurnally sampled TRMM-SLH over ocean, we consider in the following only precipitating clouds over ocean."*

  *From Fig. 5, we found that the CIRS-ML LP is more similar to the LP of the diurnally sampled TRMM-SLH over the ocean, particularly at 1:30 AM/PM. Therefore, these values could be taken as the average over the whole diurnal cycle.*

- Fig. 14: this is an interesting result that LP-ACRE relation shows no distinction between developing, mature and dissipating stages, but is dependent on MCS sizes. Not sure how to understand/explain this.

  *Indeed, we found this result also interesting, and so far we have no explanation for this, but we thought this result is important to show, so that the community may go further on this research.*

*It may be important for convective parameterizations. However, to avoid confusion, we have removed this panel from Fig. 14 in the revised manuscript.*

- Line 457: What are 'the larger ones' refer to?

*Sorry, this was a typo: What we meant to say is that larger systems have, on average, a smaller core fraction than smaller ones. This error has been fixed, and "larger" is now changed to "smaller."*

**References:**

Harrop, B. E., and D. L. Hartmann: The role of cloud radiative heating within the atmosphere on the high cloud amount and top-of-atmosphere cloud radiative effect, J. Adv. Model. Earth Syst., 8, 1391–1410, doi:10.1002/2016MS000670, 2016.

Haynes, J. M., L'Ecuyer, T. S., Stephens, G. L., Miller, S. D., Mitrescu, C., Wood, N. B., and Tanelli, S.: Rainfall retrieval over the ocean with spaceborne W-band radar, Journal of Geophysical Research: Atmospheres, 114, https://doi.org/10.1029/2008JD009973, 2009.

Li, Y., D. W. J. Thompson, and S. Bony: The Influence of Atmospheric Cloud Radiative Effects on the Large-Scale Atmospheric Circulation. J. Climate, 28, 7263–7278, https://doi.org/10.1175/JCLI-D-14-00825.1, 2015.

Stubenrauch, C. J., Caria, G., Protopapadaki, S. E., and Hemmer, F.: 3D radiative heating of tropical upper tropospheric cloud systems derived from synergistic A-Train observations and machine learning, Atmospheric Chemistry and Physics, 21, 1015–1034, https://doi.org/10.5194/acp-21-1015-2021, 2021.

Stubenrauch, C. J., Mandorli, G., and Lemaitre, E.: Convective organization and 3D structure of tropical cloud systems deduced from synergistic A-Train observations and machine learning, Atmospheric Chemistry and Physics, 23, 5867–5884, https://doi.org/10.5194/acp-23-5867-2023, 2023.

---

## Author Comment (AC2)

**Author response to Referee #2:**

Dear Referee,

We sincerely thank you for your thoughtful and constructive comments, which have greatly helped us improve the manuscript. Below, we provide our point-by-point responses to each comment. All changes made to the manuscript according to the comments of the three reviewers have been carefully highlighted in the revised version.

**General comments:**

This study investigates the latent and radiative heating fields of tropical cloud systems using synergistic satellite observations. The artificial neural network (ANN) regression is used to generate 'observational data' based on limited satellite observations and meteorological reanalyses. This work could be useful for understanding tropical cloud systems, particularly mesoscale convective systems. Overall, the paper is well-written. However, the presentation and analysis need improvement before the paper is suitable for publication.

**Recommendation: Major revision**

**Major comments:**

- Some conclusions are not well supported by the presented results. For example, in lines 515-518, it is stated that "The smaller CIRS-retrieved height may be interpreted as anvils of convective systems having descended at a later stage of their life cycle or as relatively thick clouds with diffusive tops, for which the retrieved (radiative) height may be deeper within the cloud because of very small ice water content in the upper part of the cloud." However, the authors do not provide any related analysis to support this conclusion.

  *Thank you very much for your valuable comment.*

  *Indeed, these were plausible explanations, which are not easy to prove. Therefore, we wrote* ***'may be interpreted'***. *We have provided references for the latter (diffusive cloud top) in section 5.1, lines 436-437: e.g. Liao et al. 1995; Stubenrauch et al. 2010, 2017, and we have added one reference which suggests a descend of anvils with life time (Fig. 5.15 of Strandgen 2018) in line 435. We did not repeat these references in the conclusions, but since these are only possible explanations and they are not very clear and not so important, we have removed them from the conclusions.*

  *To improve the manuscript, we have made substantial revisions to the conclusion section, as well as other parts of the text where the explanations were unclear.*

- Lines 144-145: "The rain intensity classification (no rain, light rain, heavy rain), determined by an ANN trained with precipitation data from CloudSat, considers light rain to be $< 5$ mm h$^{-1}$ and heavy rain $> 5$ mm h$^{-1}$ (Stubenrauch et al., 2023)."
  a) The GPCP and TRMM precipitation data are widely used by the community. Could you provide a justification for why you chose to use CloudSat precipitation data? Also, please include the link to the CloudSat precipitation data in the Data Availability Statement.

  *If we wanted to use information on the rain intensity or rain fraction in the training, we needed to use data which are available over the whole production period (2004-2018) for each of the AIRS and IASI footprints. Since we have used CloudSat-CALIPSO data earlier to expand the radiative heating rates (Stubenrauch et al. 2021), we used CloudSat precipitation rate data (2C-PRECIP-COLUMN, Haynes et al. 2009) to expand information on precipitation (Stubenrauch*

*et al. 2023). Since the distribution of precipitation rate is highly skewed, we were not able to do an ANN regression of this precipitation rate, but we succeeded to build an ANN classification, separating no-rain, light-rain and heavy-rain scenes (Stubenrauch et al. 2023). In this article, we have also shown that this scene identification was more powerful to separate TRMM SLH latent heating rates than cold brightness temperature.*

*We could not use TRMM data, because as we have shown, before, these are only available for less than 5% of the AIRS and IASI footprints. There may be a possibility to use GPCP data, but again the L2 data would have been needed to collocate with the complete AIRS and IASI datasets in time and space, and we are not sure if there is a 100% overlap. Since we have had already this CIRS-ML rain intensity classification, we pursued this path.*

*During this review, we have analysed the complete collocated dataset further, and we found indeed some noise linked to the difference between fraction of certain rain from CloudSat (using CIRS-ML) and from TRMM, but the CIRS-ML dataset seems to be coherent with the certain rain fraction, though it was not used as input for the ANN (see replies to referee #1).*

*In this article we have shown one method to expand the latent heat with CIRS data; there may be better ways, but since the whole procedure took quite an effort, it was not possible to pursue two paths in parallel. It would be nice if this work inspires other researchers to improve the methods to get a complete 3D dataset of UT cloud systems.*

*Thanks for your valuable comment. We have completely revised the last paragraph of section 2.3, and we have added the comparison between TRMM and CIRS-ML certain rain fraction to section 3.2..*

 b) How did you handle data when the rain rate is exactly 5 mm h$^{-1}$?

*Actually, these thresholds are valid for the CIRS-ML rain intensity classification at the AIRS footprint scale. We have now completed the description by including the propagation to the 0.5° grid cells. At the 0.5° scale, the threshold should be more around 2.5 mm/h. Since we use always the same scene identification, for training and production, the categorization should be still coherent. All thresholds are <= for light and > for heavy rain.*

- Could you please provide the definitions of UT and UT clouds in this study? How are the CRE and ACRE calculated, given that they represent the clouds' impact on radiative heating?

*UT clouds were defined in line 331 (previous version): "We consider UT clouds with $P_{cld}$ < 350 hPa..."*

*High-level clouds were defined in line 118 (previous version): "CIRS cloud types are defined according to $p_{cld}$ and $\varepsilon_{cld}$ as high clouds ($P_{cld}$ < 440 hPa)..."*

*Since the definition of UT clouds is introduced much later, it may cause confusion with high-level clouds. To address this, we added the sentence "UT clouds with $P_{cld}$ < 350 hPa are part of the high cloud category." at line 149 in revised manuscript.*

*Indeed, there was a confusion in the manuscript. The training of the ANNs was separately done for high-level clouds and mid- and low-level clouds. We have changed 'UT clouds' to 'high-level clouds' in both the figures and the text, mostly in section 2.*

*When moving to the analysis of mesoscale convective systems, we use the definition of UT clouds.*

*The cloud radiative effect (CRE) is calculated as the difference between all-sky radiative heating rates and clear-sky radiative heating rates, with the unit K/day. We use the clear sky identification of CIRS as in (Stubenrauch et al. 2021). The atmospheric cloud radiative effect (ACRE), already defined for example by Li et al. (2015) and Harrop and Hartmann (2016) as the difference in cloud radiative effects between the TOA and the surface, and corresponds to the vertically integrated CRE, with unit W/m².*

*To clarify this point, we have revised the sentence on line 61 (previous version) to:*

*"In our analyses, we use the following definitions: LH refers to the latent heating profile; LP denotes the vertically integrated latent heating; Qrad represents the radiative heating profile; CRE (cloud radiative effect) refers to the difference between all-sky and clear-sky radiative heating rates, expressed in units of K/day; and ACRE (atmospheric cloud radiative effect) represents the vertically integrated CRE, with units of W/m²."*

- Different time periods are selected for analysis (e.g., Fig. 2 (2008-2013), Fig. 3 (2004-2013/2007-2010), Fig. 9 (2004-2018)). Why were these specific time periods chosen?

*For the ANN training in Section 2 we used collocated data. While the CIRS-ML AIRS data are available from 2004 on, the CIRS-ML IASI data are available from 2008 on. Since we preferred to use complete years for the collocation, and in 2014 there were two months missing for TRMM, we used the period until 2013.*

*The CIRS-ML LH production was done over the complete periods of CIRS-AIRS (2004-2018) and CIRS-IASI (2008-2018), as explained in the beginning of section 3.*

*After verifying the coherence between original TRMM-SLH LP, including the diurnal cycle, and ML-predicted LP at the different observation times in Figure 5, we decided to use only CIRS-AIRS data over ocean for our analysis. To maximize the statistical reliability, we utilized the entire CIRS-AIRS dataset (2004-2018).*

- Line 336: "An MCS is defined as an UT cloud system with at least one convective core and the presence of precipitation." How did you distinguish between an MCS and an isolated deep convective cloud system?

*Isolated deep convective systems would be smaller. We look at a scale of grid cells of 0.5°, which leads to a size of about 2500 km². But indeed, the definition of a MCS is the scale of 100 km and larger. So, we changed it in the text to convective systems. However, for the analysis of mesoscale organization we excluded convective systems which are built of less than 5 grid cells, so these are MCSs.*

- Line 293-295: "In summary, the increasing slopes of the relationship between LP of CIRS–ML and TRMM suggest that our ML-expanded LH dataset is suitable at scales larger than about 2.5° (with a slope of 0.7)."
  a) It would be better to present a plot with the 2.5° scale in Figure 6.
  b) Line 501: "with slopes of 0.68 and 0.76 for 2.5° and 5°, respectively." Please check the slope for the 2.5° scale: is it 0.68 or 0.7?

*Thanks for your comments, we have added the plot for the 2.5° scale to Figure 6, as suggested. For the slope of 2.5° scale, you are correct that it is 0.68, not 0.7. We appreciate your careful attention to this detail, and the correction has been made in the revised manuscript.*

- Figure 11 shows negative ACRE when LP < 200 W/m² for precipitating UT clouds over the ocean. However, Figure 10b shows positive ACRE when LP < 200 W/m² for precipitating UT clouds over the ocean. Why is there a discrepancy?

  *Thank you for this thoughtful comment.*

  *Fig.10b: LP is divided into multiple bins, and the average ACRE within each bin is calculated.*

  *Fig.11: A 2D histogram (pcolormesh) is used to divide the spatial distribution of LP and ACRE into small grids, and the average value of the variable within each grid is calculated. This method captures more detailed local variations and distribution characteristics.*

  *The discrepancy is because:*

  *In Fig.10b, for LP < 200 W/m², the ACRE is positive because each bin contains multiple data points, and negative values are masked by the averaging process.*

  *In Fig.11, each small grid may contain only a few data points, allowing negative ACRE values to be displayed.*

  *Indeed, the comparison between Figures 10b and 11 shows that there are only few cases with a negative ACRE. These cases occur exclusively in cool-dry environments. In the text (lines 413-414, previous version) we explained that these should be thin cirrus with lower precipitating clouds underneath. The ACRE of these scenes leads to a slightly negative ACRE (Stubenrauch et al. 2021). The CIRS retrieval only identifies the uppermost cloud layer in the case of multi-level clouds.*

**Minor comments:**

- Line 101-102: "Its spectral coverage spans 2378 radiance channels within the wavelength range of 3.7–15.4 μm (650–2700 cm$^{-1}$)"
  What does "650–2700 cm$^{-1}$" represent?

  *The range 650–2700 cm$^{-1}$ represents the wavenumber range, which is another way to express the wavelength range 3.7–15.4 μm.*

  *Wavenumber (cm$^{-1}$) is the number of wave cycles per centimeter and is commonly used in infrared spectroscopy. It is related to wavelength (μm) by the formula:*

  Wavenumber $= 10^4$ / wavelength (μm)

  *3.7 μm corresponds to ~2700 cm$^{-1}$, and 15.4 μm corresponds to ~650 cm$^{-1}$.*

  *Since this is of no importance for our article, we took out the wave numbers. We have updated line 102 to make this clearer and avoid any confusion.*

- Line 214-215: "In all cases, the real data reveal a large variability between 600 and 800 hPa, which may be linked to the variability between stratiform and convective rain within the 0.5°." Why?

  *Depending on the fraction of convective and stratiform rain within the 0.5° grid cells, and because the stratiform and convective shapes of the LH profiles are very different (e. g. Schumacher et al. 2004), the shapes vary.*

- Line 232: "The minor cooling observed at approximately 550 hPa is attributed to the melting process" Why does melting induce cooling in the radiative longwave (LW) heating profile?

  *The cooling observed at approximately 550 hPa due to melting is a result of latent heat absorption. When solid precipitation (such as snow or ice) melts, it requires energy to transition from a solid to a liquid state. This energy is drawn from the surrounding atmosphere in the form of latent heat, which leads to a cooling effect in the local environment. In the radiative LW heating profile, this cooling appears because the absorbed energy lowers the local temperature (Yasunaga et al., 2008), reducing the thermal emission in the longwave spectrum.*

- Figure 5: Why does the vertically integrate LH still have the units of K/day? Which lines represent 9:30 AM/PM?

  *Thanks for these questions.*

  *In Figure 5, both y-axes originally showed vertically integrated latent heating (in W/m²) and averaged latent heating (in K/day), by using a conversion factor. To avoid confusion, we have removed it and kept only the W/m² axis.*

  *The new Figure 5 only shows vertically integrated LH (LP) in W/m². We have also simplified the figure for clarity. Since our main focus is the coherence between TRMM–SLH LP and ML-predicted LP, we removed details about SW and LW radiative effects and related text to reduce complexity. After the revision, Figure 5 presents zonal averages of original TRMM LP, including the diurnal cycle, and LP from ML regression using CIRS-AIRS and CIRS–IASI at four observation times of 1:30 and 9:30, each AM and PM.*

- Figure 7: "10-year (2008–2018 JAN) averages" Should it be 11 years instead?

  *Thank you for catching this. You are correct, it should be 11 years instead of 10. We have updated this in the text.*

- Figure 8: Can you show the plot of the difference between CIRS-ML and TRMM/GPCP? The title of the label bar should be LP instead of LH.

  *Figure 8 is an illustration to show that the CIRS-ML LP captures the geographical distributions. We have already shown before (Figures 3 and 6 and also new Figures in the supplement) that CIRS-ML regression leads to a smaller range of LP, meaning that small LP is slightly overestimated and larger LP may be largely overestimated. Instead of showing a difference plot in addition, we show a ratio (new Figure S13), which should reflect the slopes given in Figure 6.*

  *Thank you for your careful review. We have corrected the mistake in the title of the label bar, which has been changed to "LP" in the revised manuscript.*

- Line 348: "Figure 9 presents profiles of latent heating and radiative heating…." The label of Figure 9b indicates that the cloud radiative effect (CRE) is presented.

  *We apologize for the confusion. The plot actually shows the cloud radiative effect (CRE), which is the difference between all-sky radiative heating rates and clear-sky radiative heating rates. To clarify, we have corrected this in the text. Thank you for pointing it out.*

- Line 440: circulation Stephens et al. (2024) -> circulation (Stephens et al. 2024).

*Thank you for pointing this out. The error has been corrected to "circulation (Stephens et al., 2024)." We appreciate your attention to detail.*

- Line 405: "Humid environments increase the buoyancy of convective clouds, which allows clouds to reach greater heights (Holloway and Neelin, 2009), confirmed by Fig. 10b". However, Fig. 10b does not provide any information about cloud heights.

*We apologize for the mistake. The correct reference should be Fig. 10c, not Fig. 10b. We have corrected this in the text. Thank you for pointing it out.*

- Line 461: "20 km$^2$" -> "20 W m$^{-2}$"

*Thank you for catching this error. It has been corrected to "20 W m$^{-2}$." We appreciate your careful review.*

- "LP" is used to represent two different physical variables in one study.
  a) Line 60: "In our analyses, we use the following definitions: LH for latent heating profile, LP for vertically integrated LH"
  b) Line 453: "The relationship between precipitation intensity (LP) and radiative enhancement (ACRE)"

*Thank you for your thoughtful comment. In this study, LP is consistently defined as vertically integrated latent heat. However, we use LP as a proxy for precipitation intensity. To avoid confusion, we have clarified this usage in the revised text (line 547):*

*" ...less organized and larger, more organized MCSs at similar average rain intensity (using vertically integrated latent heating LP as a proxy).*

**References:**

Harrop, B. E., and D. L. Hartmann: The role of cloud radiative heating within the atmosphere on the high cloud amount and top-of-atmosphere cloud radiative effect, J. Adv. Model. Earth Syst., 8, 1391–1410, doi:10.1002/2016MS000670, 2016.

Liao, X., Rossow, W. B., and Rind, D.: Comparison between SAGE II and ISCCP high-level clouds: 2. Locating cloud tops, Journal of Geophysical Research: Atmospheres, 100, 1137–1147, https://doi.org/https://doi.org/10.1029/94JD02430, 1995.

Li, Y., D. W. J. Thompson, and S. Bony: The Influence of Atmospheric Cloud Radiative Effects on the Large-Scale Atmospheric Circulation. J. Climate, 28, 7263–7278, https://doi.org/10.1175/JCLI-D-14-00825.1, 2015.

Stubenrauch, C. J., Cros, S., Guignard, A., and Lamquin, N.: A 6-year global cloud climatology from the Atmospheric InfraRed Sounder AIRS and a statistical analysis in synergy with CALIPSO and CloudSat, Atmospheric Chemistry and Physics, 10, 7197–7214,https://doi.org/10.5194/acp-10-7197-2010, 2010.

Stubenrauch, C. J., Feofilov, A. G., Protopapadaki, S. E., and Armante, R.: Cloud climatologies from the infrared sounders AIRS and IASI: strengths and applications, Atmospheric Chemistry and Physics, 17, 13 625–13 644, https://doi.org/10.5194/acp-17-13625-2017, 2017.

Stubenrauch, C. J., Caria, G., Protopapadaki, S. E., and Hemmer, F.: 3D radiative heating of tropical upper tropospheric cloud systems derived from synergistic A-Train observations and machine learning, Atmospheric Chemistry and Physics, 21, 1015–1034, https://doi.org/10.5194/acp-21-1015-2021, 2021.

Stubenrauch, C. J., Mandorli, G., and Lemaitre, E.: Convective organization and 3D structure of tropical cloud systems deduced from synergistic A-Train observations and machine learning, Atmospheric Chemistry and Physics, 23, 5867–5884, https://doi.org/10.5194/acp-23-5867-2023, 2023.

Strandgren, J.: The life cycle of anvil cirrus clouds from a combination of passive and active satellite remote sensing. Dissertation, LMU München: Faculty of Physics, 2018

Yasunaga, K., Hashimoto, A., & Yoshizaki, M.: Numerical simulations of the formation of melting-layer cloud. Monthly weather review, 136(1), 223-241, 2008.

---

## Author Comment (AC3)

**Author response to Referee #3:**

Dear Referee,

We sincerely thank you for your thoughtful and constructive comments, which have greatly helped us improve the manuscript. Below, we provide our point-by-point responses to each comment. All changes made to the manuscript according to the comments of the three reviewers have been carefully highlighted in the revised version.

**General comments:**

This paper is to use synergistic satellite data from active instruments; it also applies artificial neural network regressions on InfraRed Sounder data, and meteorological reanalyses to investigate the relationship between latent heating (LH) and radiative heating (RH) for mesoscale convective systems (MCS).

The main results show (1) the zonal averages of vertically integrated LH (LP) at 1:30AM and PM LT align well with those from the diurnal sampling of TRMM–SLH LH over ocean; (2) the surface temperature has a larger impact on the **atmospheric cloud radiative effect (ACRE)** in dry than in humid environments for **Upper Tropospheric (UT)** clouds; (3) humidity plays a large role in enhanced ACRE for lower clouds, producing relatively small latent heat; (4) the mean ACRE per MCS increases with LP; and (5) LH profiles of mature MCSs have a larger contribution of stratiform rain than the smaller MCSs,

This paper is interesting because it applies artificial neural network to investigate the relationship between LH and RH.

I have comments that the authors need to address to have paper to be published.

**Specific comments**

- Line 17: Suggest deleting "the precipitating parts of".
- Line 25: Suggest deleting "closer".

  *All taken into account*

- Line 32, tropical: May identify the area of "tropical: (i.e., 30o or 20o south to north).

  *Thank you for your suggestion. We agree that specifying the exact latitudinal range of the tropical region improve clarity from the beginning on. In the revised manuscript, we have added the following sentence at the end of this paragraph:*

  *"... The data expansion and the following analyses cover the latitudinal band 30°N-30°S."*

- Line 83; Hagos et al.: Not sure if this reference needed.

  *Thank you for pointing this out. Since Hagos et al. (2010) does not directly compare the latest versions of the SLH and CSH retrievals, we have removed this part from the text.*

- Line 85, continuous: Not sure what it means.

  *"Continuous" refers to how the SLH algorithm generates vertical profiles of latent heating that smoothly vary with PR echo-top height, rather than relying on a limited set of discrete profiles stored in LUTs, as described in Hagos et al. (2010). However, since Hagos et al. (2010) does not reflect the latest versions of either SLH or CSH, we have removed this sentence. In the revised text,*

*we have rewritten this paragraph to clarify the methodologies used in the latest versions of SLH and CSH, and provided a clearer comparison between the two algorithms.*

- Line 87: Please check the year of Shige et al. (2004 or 2003).

  *We apologize for the error. The correct year should be Shige et al. (2004), and we have now corrected it in the manuscript. Thank you for catching this mistake.*

- Line 94-95: Tao et al. (2022) did not state (or show) the comparison between SLH derived heating and re-analyzed heating profile. The CSH and SLH derived LH shown in Hagos et al. (2010) are from old version of CSH and SLH retrieved LH. The new version for both SLH and CSH is V6 (shown in Tao et al. 2022).
  In addition, the reanalyzed is a combination of model and observation. That is why different re-analyzed LH (shown in Hagos el. 2010) are different.

  *Thank you for your valuable comment. In the revised manuscript, we have removed the comparison between CSH and SLH from Hagos et al. (2010), as it is based on older versions of both algorithms. Instead, we have carefully reviewed and incorporated the comparison between the latest versions of CSH and SLH (V6) presented in Tao et al. (2022). The related text has been revised accordingly.*

- Line 95: It is Shige et al. (2007) that paper described the SLH algorithm.

  *Thanks for your comment. You are right, Shige et al. (2007) provides a more detailed description of the SLH algorithm. We have updated the citation in the revised manuscript to better reflect this.*

- Line 99-128 (section 2.2): What is the horizontal and vertical resolution ($0.5 \times 0.5$ degree and 50 layers) of these three-satellite derived cloud information (AIRS, IASI and CIRS)?

  *First of all, AIRS and IASI are IR sounder instruments. CIRS is the cloud retrieval algorithm which is applied to both of them. Since sounders are passive instruments, the cloud retrieval only provides information on the uppermost cloud in the case of multi-layer clouds. Therefore, there is only a horizontal resolution, because the results are 2D, with latitude and longitude as coordinates. The initial spatial resolution of these instruments is 13.5 km and 12 km at nadir, respectively (lines 100-110, previous version). Then these data have been merged to 0.5°, but with keeping sub-grid information (coming from the footprints), like fraction of different cloud types and rain fractions shown in Table 1. The vertical resolution of the CIRS-ML radiative heating rates, which were initially derived from CloudSat-CALIPSO data (Stubenrauch et al. 2021), has been reduced to 21 layers, because the input information is only 2D.*

- ERA interim's horizontal resolution is $0.75 \times 0.75$ degree. Could you please describe what are spatial resolution of these three-satellite derived cloud information? Does cubic spline function also apply to the satellite cloud information ($0.5 \times 0.5$ or $075 \times 0.75$ degree)?

  *The cubic spline interpolation was only used for time interpolation. ERA-Interim data were used as ancillary data (for T, humidity profiles and surface T and pressure as well as snow / ice presence) in the CIRS retrieval (Stubenrauch et al. 2017). The collocation in space was such that one array of footprints ($3 \times 3$ for AIRS and $2 \times 2$ for IASI) was assigned to the closest ERA-Interim grid cell.*

  *To improve clarity, we have added the corresponding description in the revised manuscript.*

- Line 135: What is sub-grid structure? Is it for making all data to $0.5 \times 0.5$ degree and 10 vertical layers?

*The subgrid structure comes from the footprints which have a size of about 13.5 km and 12 km at nadir for AIRS and IASI, respectively, and therefore is only the horizontal subgrid structure. We added this in the text to make it clearer.*

*The vertical structure of the ERA-Interim T and water vapour profiles used for the ANN regressions was reduced to 10 layers (section 2.3, line 163-165).*

- Line 144-145: Would it be nice to also use TRMM/GPM derived rainfall intensity?

*If we wanted to use information on the rain intensity or rain fraction in the training, we needed to use data which are available over the whole production period (2004-2018) for each of the AIRS and IASI footprints. Since we have used CloudSat-CALIPSO data earlier to expand the radiative heating rates (Stubenrauch et al. 2021), we used CloudSat precipitation rate data (2C-PRECIP-COLUMN, Haynes et al. 2009) to expand information on precipitation (Stubenrauch et al. 2023). Since the distribution of precipitation rate is highly skewed, we were not able to do an ANN regression of this precipitation rate, but we succeeded to build an ANN classification, separating no-rain, light-rain and heavy-rain scenes (Stubenrauch et al. 2023). In that article, we have also shown that this scene identification was more powerful to separate TRMM SLH latent heating rates than cold brightness temperature.*

*We could not use TRMM data, because as we have shown before, these are only available for less than 5% of the AIRS and IASI footprints. There may be a possibility to use GPCP data, but again the L2 data would have been needed to collocate with the complete AIRS and IASI datasets in time and space, and we are not sure if there is a 100% overlap. Since we have had already this CIRS-ML rain intensity classification, we pursued this path.*

*During this review, we have analysed the complete collocated dataset further, and we found indeed some noise linked to the difference between fraction of certain rain from CloudSat (using CIRS-ML) and from TRMM, but the CIRS-ML dataset seems to be coherent with the certain rain fraction, though it was not used as input for the ANN (see replies to referee #1).*

*In this article we have shown one method to expand the latent heat with CIRS data; there may be better ways, but since the whole procedure took quite an effort, it was not possible to pursue two paths in parallel. It would be nice if this work inspires other researchers to improve the methods to get a complete 3D dataset of UT cloud systems.*

- Figure 1: What do dark, and light blue color represent?

*We apologize for the lack of clarity in the original figure caption.*

*Dark and light blue color represent the time difference between TRMM and AIRS. We have added the colorbar and one sentence in caption of Figure 1 to:*

*"The narrow swaths represent TRMM–PR orbits, while the broader swaths represent Aqua–AIRS orbits. Shades of blue indicate variations in sampling time difference."*

- Line 160: What do you mean the "maximum" 0.0 mm h-1?

*Indeed, this was not clear. We meant a large peak at 0.0 mm $h^{-1}$. To clarify, we have revised this in the text to "with a peak at..."*

- Line 162-166: The "%" is for frequency (or area coverage). The non-precipitation means no surface rain. Correct?

*Yes, it's correct, the "%" is for frequency. And the non-precipitation means no surface rain.*

- Figure 2: The LH profiles are from SLH algorithm. Correct? In addition, please just refer one specific SLH paper that describes the algorithm design. The figure uses two different scale for LH (from -10 to 40) and RH (-10 to 30). Suggest using the same scale for both LH and RH.

  *Yes, the LH profiles are derived from TRMM-SLH data collocated with CIRS-AIRS and CIRS-IASI (see section 2.4).*

  *We have also revised the caption to reference only Shige et al. (2007) to describe the algorithm design, as you suggested.*

  *In addition, we have adjusted the scales in Figure 2 to ensure consistency between LH over ocean and LH over land.*

  *We sincerely thank the reviewer for these valuable suggestions, which have helped us improve the clarity and consistency of the figure and the manuscript.*

- Line 160-211; section 2.5.1: It may be a good idea to have a schematic diagram that shows the design of ANN for predicting LH/RH. Maybe a good idea to show the key parameters for predicting LH in diagram.

  *We appreciate the reviewer's helpful suggestion. A schematic diagram illustrating the structure of the ANN used in this has been added as Figure S2 in the supplementary material.*

  *Unfortunately, we could not find out the key parameters. We tried this already for the expansion of the radiative heating rates, but we only found that once the basic 20 variables were used, the improvement with any additional variable was small (Stubenrauch et al. 2021). We have shown all variables used for the training in Table 1.*

- Would ANN-LH predict certainty or uncertainty?

  *The ANN-LH model currently provides deterministic predictions, meaning it outputs a single value for latent heat without explicitly quantifying uncertainty. However, we acknowledge the importance of uncertainty estimation in predictive modeling. In order to get an idea of the uncertainties and in particular of the biases, we have done the analyses described in section 3.2 and 3.3, unfortunately most are only qualitative, but they give an idea. Since Figure 3 shows that the range of the predicted LH is reduced compared to the original data, the slopes in Figure 6 show that in general extreme values are not predicted. Furthermore, for small LP the predicted LH is larger when there is a mismatch between TRMM rain fraction and CIRS-ML rain fraction (originally from CloudSat), which may come from different sampling sizes (5km against 1.25 km × 2.5 km).*

- Line 185: What is impact on "randomly divided" on the retrieved product? Line 186: What is the validation data? (Is it SLH derived LH?)

  *The random division of the collocated dataset into training, validation, and test sets ensures that the data distribution is statistically similar across all subsets.*

  *Yes, the validation data is SLH-derived LH from 20% of the total input data. It is used for iterative optimization during model development, helping to tune hyperparameters and prevent overfitting.*

- Line 215: How are the convective and stratiform rain classified (from TRMM or else)?

*In the beginning, we looked at the stratiform and convective parts in the TRMM SLH data, but we did not keep this information for further study, because we found that at a spatial resolution of 0.5°, if the convective LH is very strong, the shape is dominated by this. However, since the pure stratiform and convective shapes of the profiles are very different (see for example Chang and L'Ecuyer 2019, Figure 3), the variability mostly comes from the partition of these (and additional shallow convection).*

- Line 223: What is the "true data"?

*We sincerely apologize for the confusion caused by the inconsistent terminology used in the manuscript.*

*In this case, the true data is the same as the test data, and it comes from the original TRMM-SLH data. It is a completely independent dataset that is never seen during model training and validation. It is only used at the end to evaluate the model's final performance.*

*To avoid further misunderstanding, we have revised the text to:*

*"We have evaluated the LH prediction results by comparing them with those of the target TRMM-SLH, using the 20% test data of the collocated data.*

- Figure 3: RH is for longwave (not total radiative heating/cooling). Why is only LW shown? Line 230: How do the meting processes affect longwave cooling?

*In Figure 3, the choice of LW was made it provides a clear and representative illustration of the radiative processes we aim to analyze. While we acknowledge that total radiative heating (including both LW and shortwave components) could also be considered, we believe that the LW component alone is sufficient to demonstrate the key features and validate the model's performance. We were here interested in the variability of the profiles. There exists a whole other article which shows the performance of the CIRS-ML expansion of the radiative heating, which includes the SW part (Stubenrauch et al. 2021).*

*The cooling observed at approximately 550 hPa due to melting is a result of latent heat absorption. When solid precipitation (such as snow or ice) melts, it requires energy to transition from a solid to a liquid state. This energy is drawn from the surrounding atmosphere in the form of latent heat, which leads to a cooling effect in the local environment (Yasunaga et al., 2008). In the radiative LW heating profile, this cooling appears because the absorbed energy lowers the local temperature, reducing the thermal emission in the longwave spectrum.*

- Figs. 2, 3 and 4: Please also plot/show total and UT LH in Figure 3 (for comparison with LH showing in Figure 4). In addition, please plot/show the LH over ocean vs land as those shown in Figure 4.

*Thank you very much for the suggestion. Indeed, we show Figure 3 separately over ocean and over land in the supplement (Figure S7). We had initially also Figure 4 separately over ocean and land for TRMM-SLH in the supplement, but then we tried to reduce the number of figures. So, we have put this figure back (now Figure S1), and we have also done Figure 3 for all and UT clouds, over ocean and land, as you suggested (Figure S5). In the text we have added the comparison between these.*

- Line 267: Change minor to small (also in other places).

*Taken into account*

- Figure 5: What is LP from TRMM? Is it SLH derived LH? Is GPM data used?

  *Yes, LP from TRMM is the vertically integral of the SLH profiles. Only TRMM data are used.*

- Please use the term TRMM-LSH (not TRMM). Please also change TRMM to TRMM-LSH in the text and figure caption.

  *Yes, it is SLH derived LP, we have changed TRMM to TRMM-SLH in the text and figure caption. Thank you for your thoughtful suggestion.*

- Fig.6: Please elaborate why LH/RH is only from 10 S to 10 N). Why does not show, 30S – 30, as shown in other figures?

  *Thank you for your comment. The intention here was to illustrate the non-uniform diurnal sampling issue of TRMM. To simplify the manuscript and avoid adding too many figures, we chose to show only the results for 10°S–10°N, as this region has a shorter TRMM repeat cycle, leading to more reliable results: the slopes between monthly mean LP of TRMM-SLH and CIRS–ML are 0.54 to 0.82 in the band close to the equator (10°N-10°S), they vary from 0.44 to 0.77(see figure below) at the higher latitudes (20°-30° N/S) with a TRMM repeat cycle which is only half as large, we put here the figures correspond to Fig.6 for 20°-30°N/S, which are not shown in the manuscript:*

[Figure]

*We have added an explanation in the revised manuscript to clarify why only the 10°N–10°S region is shown:*

*"As the TRMM revisit cycle depends strongly on latitude (Negri et al. 2002), with 23 days at the equator and up to 46 days at the highest latitudes (the latter should have different observation times sampled in different months), we limited the latitudinal band to 10°N–10°S for this comparison…"*

- Line 292-293: Please elaborate more details (why ANN does not capture extreme event; also, what is "well" extreme events). Also need to use TRMM-SLH (not TRMM) in the statement

  *It is known that ANN regression captures well the average, as was shown here. While for the radiative heating rates, the spread is also well predicted, for the LH the spread is reduced; there are several reasons for this: (1) the input information is not enough to predict the relationship precisely enough and most (2) the distributions are skewed, with most of the events have no rain and only very few have heavy rain; we could already reduce the underestimation by separately training for different rain intensity scenes. We have shown in (Stubenrauch et al. 2021), that by training over all high-level clouds, the average of the heating rates was smaller than when separately training for Cbs, which are rare.*

  *To make it clear, we have added one sentence in the revised manuscript:*

  *"... the ANN regression does not capture well extreme events, mainly because of insufficient input information and the skewed distribution of input data, as mentioned in section 2.5…"*

  *And we have corrected all occurrences to consistently use TRMM-SLH instead of TRMM where appropriate. Thank you for helping us improve the clarity and precision of the text.*

- Line 298: Change Figure 7a-d to Figure**s** 7a-d (or change show to shows).

  *Taken into account*

- Line 312-313: Why is there "less convective activity, there is more low-level cloud formation"? Figure 7: What is 1h 30 AM/PM? (at the end of caption).

  *When convection is weak, vertical air movement is limited, making the atmosphere more stable. This causes moisture to build up in the lower atmosphere, which helps form low-level clouds.*

  *1:30 AM/PM is the sampling time of AIRS. This is to show that the LP data we use for this figure comes from ML regression trained on TRMM-AIRS collocated data. To avoid confusion, we changed the end of the Figure 7 caption to: "...at 1:30 AM/PM (AIRS)"*

- Figs. 7 and 8: Suggest quantifying the similarity and difference with statistical analyses.

  *We sincerely thank the reviewer for this valuable suggestion. These figures were primarily intended for illustration purposes, to provide context for the overall conditions during the selected La Niña and El Niño periods. The main focus of our analysis is presented in Section 5. There have been many publications on ENSO, and considering that ENSO itself is not the primary focus of this study, we believe that a more detailed statistical analysis on these particular figures would not significantly enhance the scientific contribution of the paper, as it would not add substantial new knowledge regarding ENSO.*

  *However, we have now provided the ratios (CIRS–ML LP / TRMM LP or GPCP LP) corresponding to Figure 8 and have added a discussion on these ratios in the revised manuscript.*

- Line 343: What is "less" 26,000 MCSs? Please also show the global – geographic distribution of these MCSs. Are only convective cores in these MCSs considered? Usually, there are stratiform cloud (generally with large area coverage than that of convective core). Is it possible to estimate stratiform % in these MCSs??

  *We rewrote the definition of oceanic systems and explained why we used only 25N-25S as:*

*"Furthermore, we concentrate only on oceanic systems, defined as systems with less than 10% of their size overlapping land, and we limit the latitude band to 25N-25S, because most of these systems are located there according to Figure 5 of (Protopapadaki et al. 2017). These criteria leave us with about 26358 convective systems (CSs) for the period of 2004–2018."*

*The MCSs include both, convective cores and anvils. Our concept of UT cloud system reconstruction is not new, we have first presented it in (Protopapadaki et al. 2017), which also show geographical distributions (Figure 5). Then we have refined it and shown results of core and anvil properties in (for ex. Figure 7 and 8 of Stubenrauch et al. 2023). We have added a few lines into section 4 about this. Instead of determining the % of stratiform anvil, we have determined the % of the core within these systems (core fraction), which we used as a proxy for the life stage (close to one in the developing phase and decreasing with life time). This concept has been shown in the earlier publications, and it has also been used for the evaluation of a new ice scheme in the LMDZ climate model (Stubenrauch et al. 2019).*

*To help improve clarity, we have updated Section 4 in the revised manuscript by improving the explanation and adding relevant references:*

*"We define a convective system as an UT cloud system with at least one convective core and the presence of precipitation. Earlier studies (Protopapadaki et al. 2017, Stubenrauch et al. 2019, Stubenrauch et al. 2023) have shown that the convective core fraction, given by the ratio of convective core size over MCS size, can be used as a proxy of the maturity stage (e. g. Stubenrauch et al. 2023)."*

- Figure 9: Why is there large/small cooling within the Figure 9a?

  *Section 4 was brought in to describe the construction of the mesoscale convective systems. We then thought to illustrate the effect of different proxies for precipitation intensity. Some of these proxies are very indirect, like the size of the systems or the minimum cooling above the precipitating part of the convective system (the LW cooling above the cloud stands for the opacity). For the latter, we select the grid cell within the precipitating part of the convective system with the minimum value of the cooling above the cloud. We compare the latent heating rate profiles with the one for heavy rain MCSs, which is the most direct proxy. Figure 9 was meant to test the coherence of the data. The LW cooling above the core comes from the CIRS-ML data and the core identification from CIRS and using CIRS-ML rain classification. And we can show that the CIRS-ML latent heating profiles change in coherence with the different proxies.*

  *Furthermore, we show that on average the minimum LW cooling above the heavily raining MCS's is the largest. This shows again that the minimum cooling above the participating part of the convective system seems to be an interesting proxy for rain intensity.*

- Line 352, -7.5 K Day$^{-1}$; Line 356, 40 K Day$^{-1}$: It is supposed to heating (release heating from condensation/deposition) in the convective core.

  *The -7.5 K/day is the threshold of the minimum cooling above the cloud, whereas the 40 K/day is the LH. Indeed, this paragraph was not well written, we have modified it (in particular taking out the description of the thresholds which cut the flow and moving this to the legend) and hope that it is now easier to read.*

- Line 357: Change "produce" to "show".

  *Taken into account*

- Line 363-364: It is not clear, what is **intensity** is directly related to heavy precipitation? What is

heavy precipitation (do you mean precipitation event)?

*One proxy for precipitation intensity may be given by the presence of heavy precipitation. The presence of heavy precipitation means a convective system which includes at least one grid cell with a large precipitation rate (more than approximately 2.5mm/h). We have added a more detailed on the determination of light and heavy precipitation in section 2.3, and we have slightly rewritten the two paragraphs following Figure 9.*

- Line 398: Change "produce" to "release". Line 405: Change "greater" to higher.

  *All taken into account*

- Line 404-409: There are other dynamic factors "i.e., low-level wind share, CAPE" that can play important role on cloud development.

  *The impact of lower tropospheric moisture on buoyancy through entrainment appears to be particularly important compared to other mechanisms. However, you are absolutely right that additional processes may also play a role, such as CAPE and low-level wind shear, as you mentioned. Therefore, we have added a discussion of these factors in the revised manuscript.*

- Line 412: Does humidity have impact on atmospheric radiative effect? (Heat atmosphere could reduce relative humidity).

  *Here we investigated how the ACRE and LP are distributed in environments of different SST and CWV. We are not able to study feedbacks like you suggest. For this we would need a Lagrangian study, which is foreseen for future studies.*

- Line 418, 420: What is mid-heavy, bottom-heavy, and top-heavy convective regime? Line 419: Where is "on top of lower convection"?

  *Mid-heavy, bottom-heavy, and top-heavy convective regimes have been defined by Masunaga and Takahashi (2024) based on the net moisture and moist static energy (MSE) transport associated with vertical motion. Figure 5 of their article shows a very nice summary illustration. In Figure 6 of their article, they associated the different parts of the ACRE–LP space to these convective regimes, again according to import and export of net moisture and MSE.*

  *For clarity, we have revised lines 415–417(previous version) to:*

  *"The occurrences in the LP–ACRE plane and their associated average environments can be compared to the results of Masunaga and Takahashi (2024): They have characterized three convective regimes (bottom-heavy, mid-heavy, and top-heavy) based on the net moisture and moist static energy (MSE) transport associated with vertical motion (their Figure 5) and then associated different parts of the ACRE–LP space to these convective regimes, again according to import and export of net moisture and MSE (their Figure 6)."*

  *'UT clouds on top of lower convection' means 'UT clouds above lower convection'. Changed in the text.*

- Line 423, two LP intervals: Need to mention that "the two regimes are with LP > and < 500 W m$^2$.]".

  *Taken into account*

- Line 424-425: Please elaborate in detail on the statement: the LH profiles seem to be dominated by

stratiform rain, with a relatively narrow LH peak around 410 hPa. This maybe the 1ˢᵗ time that the authors mention the stratiform rain.

*We have added the following sentences in section 2.1 after the description of what the TRMM radar measures:*

*"In general, convective LH profiles show heating throughout the vertical layers, except near the surface due to evaporation at lower levels. LH profiles of the stratiform rain within the anvils show cooling at low levels below a melting level and heating at levels above (e. g. Figure 3, Chang and L'Ecuyer 2019). The peak in LH of isolated convection is also lower in altitude than the one of complex convective systems including stratiform rain in the anvils (e. g. Hartmann et al. 1984)."*

- Figure 12: The scales used in Figure 12a (-2 to 6) and Fig.12b (-5 to 25) are quite different. Please mention this in the text.

*Thank you for pointing this out. We have added the following sentence on line 429:*

*"Note that the scales of LH in Figure 12a (-2 to 6 K day−1) and Figure 12b (-5 to 25 K day−1) are different, reflecting the large difference in the LH peak values between the two LP intervals."*

- Line 436-438, life cycle: Can you justify the discussions on "can ANN produce the life cycle information"?

*Sorry, we forgot to add a reference to this finding:*

*"The smaller height may be interpreted as anvils of convective systems having descended at a later stage of their life cycle (Figure 5.15 of Strandgren, 2018) ..."*

*It was a tracking study using geostationary passive data (SEVIRI) combined with active data (CALIPSO) and ANN.*

- Figs. 13 and 14 caption: Is 1:30 AM or PM local time? Also, why not consider 30 S to 30 N (as some of other figures) in the discussion?

*Yes, it is 1:30 AM/PM local time. To make this clearer, we have revised the captions for Figure 13 to Figure 15 to say "1:30 AM/PM local time."*

*Indeed, we did not explain the reason. We expect MCSs in the deeper tropics (see geographical distribution of convective systems in Figure 5 of Protopapadaki et al. 2017), so we limited the latitude band to 25N-25S for the MCS analysis. We have added a sentence on this in the revised text :*

*"Furthermore... and limit the latitude band to 25° N-25° S, because most of these systems are located there according to Figure 5 of (Protopapadaki et al. 2017)."*

- The definition of developing, mature and dissipating stage of MCS need to be elaborated in detail (or refer to observation or show the structure – both vertical and horizontal for these)

*As mentioned before, the concept of core fraction within convective system as proxy for life stage was already presented and tested in Protopapadaki et al. 2017 and Stubenrauch et al. 2023. We added some more explaining text, pointing to figures in these publications.*

- Line 449: What is the "minimum" temperature within the convective core?

*We compare the cloud top temperature of all grid cells being part of the convective core and choose the minimum value. This variable was already used in (Protopapadaki et al. 2017) as a proxy for convective depth.*

*To clarify this point, we have revised the text as follows:*

*"These behaviors are in line with those of the fraction of precipitation area within the MCS and the minimum cloud top temperature within the convective core, respectively…"*

- Line 456, 0.35 compared to 0.60; Both are classified as mature stage. Why do you need to compare these two mature stages?

  *Figure 14b includes all MCSs, while Figure 14c only shows the result for the mature state. We wanted to show that if we compare MCSs during the same life stage, the effect becomes even clearer.*

  *To avoid unnecessary confusion, we have removed Fig 14b in the revised manuscript and now focus only on the analysis during the mature stage (Figure 14c).*

- Line 461, 20 km²: I thought 0.5 degree is horizontal resolution. Where is 20 km² for small rainfall intensity from?

  *Sorry, it was a mistake. I used the wrong unit when preparing the LaTeX manuscript. It should be 20 W/m², not 20 km². This error has been corrected in the revised version. Thanks for your question.*

- Figure 15: What is the LP 212, LP 648, LP 1513, LP 255, LP 682 and LP 1357 (within the figure)? No discussion

  *The LP intervals for which the profiles are shown in Figure 15 correspond to the first, third and fifth interval used in Figure 14. While the average LP intervals for small and large MCSs within the same interval show slight differences (LP 212 & 255 W/m², LP 648 & 682 W/m², LP 1513 & 1357 W/m², respectively for small and large MCSs), they generally correspond to approximately 220, 660, and 1400 W/m², respectively. Additionally, in response to Referee 1's concern that the results for small MCSs may be biased, we have revised both Figure 14 and Figure 15. Specifically, we have excluded MCSs smaller than 1° and, instead of applying a fixed size threshold, we now classify MCSs based on the 40% largest and smallest MCSs.*

  *We agree that this was not clearly explained in the original manuscript. To make this clearer, we have added the following explanation in the figure capture:*

  *"The LP intervals shown in the legend of Figure 15 correspond to the first, third, and fifth of Figure 14."*

  *Thank you for pointing out this omission, and we hope the added explanation clarifies the figure.*

- Line 471-473 and Figure 15(a): Please elaborate the followings
  "Larger, more organized MCSs have a larger contribution of stratiform rain than the smaller MCSs, except for the most precipitating ones which show a large heating through the whole atmosphere". It is not clear where this information from Figure 15(a)

  *If comparing with LH profiles of pure convection and stratiform anvil rain (for example shown in Figure 3 of Chang and L'Ecuyer 2019), with the stratiform having a peak which is slightly higher in altitude and a negative part below the melting point (about 5.5 km in the tropics), the LH profiles averaged over the mature MCSs with larger size show a larger decrease in the profile between peak*

*value and surface than the smaller MCSs, which can be interpreted as a larger contribution of stratiform rain. We have adapted the text in order to make it clearer.*

**References:**

Chang, K.-W., & L'Ecuyer, T.: Role of latent heating vertical distribution in the formation of the tropical cold trap. Journal of Geophysical Research: Atmospheres, 124, 7836–7851. https://doi.org/10.1029/2018JD030194, 2019.

Harrop, B. E., and D. L. Hartmann: The role of cloud radiative heating within the atmosphere on the high cloud amount and top-of-atmosphere cloud radiative effect, J. Adv. Model. Earth Syst., 8, 1391–1410, doi:10.1002/2016MS000670, 2016.

Hartmann, D. L., Hendon, H. H., and Houze, R. A.: Some Implications of the Mesoscale Circulations in Tropical Cloud Clusters for Large-Scale Dynamics and Climate, Journal of Atmospheric Sciences, 41, 113 – 121, https://doi.org/10.1175/1520-0469(1984)041<0113:SIOTMC>2.0.CO;2, 1984.

Haynes, J. M., L'Ecuyer, T. S., Stephens, G. L., Miller, S. D., Mitrescu, C., Wood, N. B., and Tanelli, S.: Rainfall retrieval over the ocean with spaceborne W-band radar, Journal of Geophysical Research: Atmospheres, 114, https://doi.org/10.1029/2008JD009973, 2009.

Masunaga, H. and Takahashi, H.: The Energetics of the Lagrangian Evolution of Tropical Convective Systems, Journal of the Atmospheric Sciences, 81, 783 – 799, https://doi.org/10.1175/JAS-D-23-0141.1, 2024.

Protopapadaki, S. E., Stubenrauch, C. J., and Feofilov, A. G.: Upper tropospheric cloud systems derived from IR sounders: properties of cirrus anvils in the tropics, Atmospheric Chemistry and Physics, 17, 3845–3859, https://doi.org/10.5194/acp-17-3845-2017, 2017.

Shige, S., Takayabu, Y. N., Tao, W.-K., and Johnson, D. E.: Spectral retrieval of latent heating profiles from TRMM PR data. Part I: Development of a model-based algorithm, Journal of applied meteorology, 43, 1095–1113, https://doi.org/10.1175/1520-0450(2004)043<1095:SROLHP>2.0.CO;2, 2004.

Shige, S., Takayabu, Y. N., Tao, W.-K., and Shie, C.-L.: Spectral retrieval of latent heating profiles from TRMM PR data. Part II: Algorithm improvement and heating estimates over tropical ocean regions, Journal of applied Meteorology and Climatology, 46, 1098–1124, https://doi.org/10.1175/JAM2510.1, 2007.

Stubenrauch, C. J., Cros, S., Guignard, A., and Lamquin, N.: A 6-year global cloud climatology from the Atmospheric InfraRed Sounder AIRS and a statistical analysis in synergy with CALIPSO and CloudSat, Atmospheric Chemistry and Physics, 10, 7197–7214,https://doi.org/10.5194/acp-10-7197-2010, 2010.

Stubenrauch, C. J., Feofilov, A. G., Protopapadaki, S. E., and Armante, R.: Cloud climatologies from the infrared sounders AIRS and IASI: strengths and applications, Atmospheric Chemistry and Physics, 17, 13 625–13 644, https://doi.org/10.5194/acp-17-13625-2017, 2017.

Stubenrauch, C. J., Bonazzola ,Marine., Protopapadaki, S. E, Musat, I.: New Cloud System Metrics to Assess Bulk Ice Cloud Schemes in a GCM. Journal of Advances in Modeling Earth Systems, 10.1029/2019MS001642, 2019.

Stubenrauch, C. J., Caria, G., Protopapadaki, S. E., and Hemmer, F.: 3D radiative heating of tropical upper tropospheric cloud systems derived from synergistic A-Train observations and machine learning, Atmospheric Chemistry and Physics, 21, 1015–1034, https://doi.org/10.5194/acp-21-1015-2021, 2021.

Stubenrauch, C. J., Mandorli, G., and Lemaitre, E.: Convective organization and 3D structure of tropical cloud systems deduced from synergistic A-Train observations and machine learning, Atmospheric Chemistry and Physics, 23, 5867–5884, https://doi.org/10.5194/acp-23-5867-2023, 2023.

Tao, W.-K., LangG, S., Iguchi, T., and Song, Y.: Goddard Latent Heating Retrieval Algorithm for TRMM and GPM, Journal of the Meteoro logical Society of Japan. Ser. II, 100, 293–320, https://doi.org/10.2151/jmsj.2022-015, 2022.

Yasunaga, K., Hashimoto, A., & Yoshizaki, M.: Numerical simulations of the formation of melting-layer cloud. Monthly weather review, 136(1), 223-241, 2008.

---

## Author Response (AR2)

**Subject:** Revised Manuscript Submission & Affiliation Clarification

Dear Editor/Reviewers,

Thank you for your email and the positive decision regarding our manuscript *"Relationship between latent and radiative heating fields of Tropical cloud systems using synergistic satellite observations"* (ID: egusphere-2024-3434). We sincerely appreciate the reviewers' constructive feedback and your editorial support throughout the review process.

As requested, we have added the CloudSat precipitation data link to the "Data Availability Statement" in the revised manuscript. The updated version has been uploaded to the submission system, and we confirm that no other changes were made. Please let us know if further adjustments are needed.

Regarding the ROR database comment, the affiliation listed in our manuscript is accurate: *"Laboratoire de Météorologie Dynamique/Institut Pierre-Simon Laplace (LMD/IPSL), Sorbonne Université, Ecole Polytechnique, CNRS, Paris, France."*

During submission, the system's ROR database only offered *"Laboratoire de Météorologie Dynamique (Palaiseau, France)"* as an option. We selected this as the closest match, because the primary institutional address for LMD is in Palaiseau, while the authors work at LMD in Paris (LMD has three different sites).

We are honored to publish in *Atmospheric Chemistry and Physics* and look forward to contributing to the scientific community. Thank you again for your time and consideration.

Best regards,
Xiaoting Chen